

# Acoustic Emission investigation for avalanche formation and release: A case study of dry-slab avalanche event in Great Himalaya

Jagdish Kapil[1], Sakshi Sharma[1,2], Karmjit Singh[1], Jangvir Singh Shahi[2], and Rama Arora[3]

[1]RDC- Snow and Avalanche Study Establishment, Chandigarh (UT) -160036, India
[2]Department of Physics, Panjab University, Chandigarh (UT) -160014, India
[3]Department of Physics, PGGC, Sector 11, Chandigarh (UT)160011, India

**Correspondence:** Sakshi (sakshisharma10@yahoo.com)

**Abstract.** Non-invasive monitoring of avalanche formation and release processes, through the use of Acoustic Emission (AE) technique, has been a research challenge since long time. In present investigation AE technique is implemented to monitor the avalanche formation and release processes through a case study of a natural avalanche event reported in Great Himalaya. The specialized AE sensor-arrestor arrays, established over the avalanche starting zone, in conjunction to a high speed multichannel

AE acquisition system have successfully recorded the avalanche event passed through the course of instability development followed by release of avalanche. A new method is devised to compute the AE based instability index, and same has been applied to quantify the instability levels of a snowpack. The prominent AE parameters and instability indices are analyzed for different window scales with respect to different AE sensors. The effect of nivological and meteorological conditions and pit analyses collected during the avalanche formation process is also discussed. The critical instability was triggered

possibly due to the excessive loading (during snowfall) of an unstable snowpack consisting of persistent weak layers which led to the avalanche release. An abnormal and abrupt increase in the AE activity was observed prior to the avalanche release. The increasing trends in instability indices have shown a good correlation to the avalanche formation and a sharp jump in instability index is attributed to a particular transition occurring across two different instability states of the snowpack. Thus, five conceptual states of snowpack are identified for instability evolution corresponding to four different transitions during

avalanche formation and release processes.

## 1   Introduction

The mechanism of avalanche formation on a slope defines the probability of an avalanche occurrence, therefore it is essential to understand the avalanche formation process through different phases of the snowpack instability development. Various nat-

ural instabilities driven by external loading or gravity pulls may result into the hazards like avalanche which poses threats to the human lives and infrastacture. The mechanism of initiation of instability within a snowpack is essential to understand the avalanche hazards (Lackinger, 1987) and accurate prediction of the avalanche release still remains a challenge. The instability





development is a process in which a system may undergo a gradual or an abrupt transition from a stable (quiescent) to an
unstable state, which ultimately may lead to a catastrophic failure or rupture of the structure (Sornette, 2002). The instabilities

developing within a snowpack could be due to spontaneous failure activities occurring within it from microscopic to macro-
scopic scales. The release of a slab avalanche depends on the types of instabilities within the snowpack (McClung, 2002).
The old snow instabilities occur due to failure of buried persistent weak layer composed of kinetic growth crystals; however,
the new snow instabilities occur due to overloading of snowpack by precipitation during a storm (Schweizer et al., 2003). A
triggering mechanism along with the presence of a weak layer is necessary condition for development of snowpack instability.

The crack propagation propensity within weak layer of a slab is presented by Schweizer et al. (2016) and later, Gaume and
Reuter (2017) have assessed the snow instability through the numerical simulations for initiation and propagation of a crack.
The instabilities built within a snowpack due to overloading from rapid precipitation, during snowfall, and presence of weak
layer are some factors responsible for release of a dry-slab avalanche (Schweizer et al., 2003). A process-based approach has
been proposed by Reuter et al. (2015) for estimation of point instability of a snowpack. The snowpack instability information

is a useful input to the avalanche forecast model (Reuter et al., 2015) but the stability data for a snowpack on avalanche slope
is very crucial and scarcely available (Schweizer and van Herwijnen, 2013; Conlan et al., 2016). For real-time estimation of
snowpack stability on an avalanche terrain, no method is available (Reuter et al., 2015). However, for prediction of avalanche
release or any imminent failure of a slab, direct assessment of snowpack instability has been a research challenge since long
time.

The acoustic emission (AE) technique could be a promising way towards monitoring and evaluation of the structural failure
(Miguel et al., 2001; Michlmayr et al., 2012; Maillet et al., 2015; Deschanel et al., 2017; Bhuiyan et al., 2017) and can provide
vital information about the progressing damage processes (Michlmayr et al., 2012). For passive and non-invasive assessment
of snowpack stability, the AE technique could be very useful. The previous research carried out on snow shows that the AE
are generated as a result of crystal dislocations and inter crystal fracturing in snow (St. Lawrence et al., 1973; St. Lawrence

and Bradely, 1977) and could be a quantitative indicator of the creep rupture or small fractures taking place at grain boundary
of snow (St. Lawrence; 1980). The ongoing failure activities within a snowpack, occurring from microscopic to macroscopic
scales, may lead to the development of instabilities prior to an avalanche release (St. Lawrence, 1980; Sommerfeld, 1982)
and the quiescent periods preceding an avalanche may be associated to the evolution of snowpack instability (Bowles and St.
Lawrence, 1977). Under natural environment, the spontaneously occurring deformation and damage processes within a snow-

pack may release AE and is very sensitive to the deformation rates (Kapil et al., 2014) and the imminent failures (Reiweger
et al., 2015). Further, to study the structural stability of geological systems, the exponent of power law distribution (b-value)
derived from AE, has been used as an indicator of failure (Amorese et al., 2009; Amitrano, 2010; Michlmayr et al., 2012; Mar-
zocchi et al., 2003). For lab-based monitoring of the failures within snow, b-value has also been studied (Reiweger et al., 2015;
Datt et al., 2015). An abrupt change in the b-value indicates a transition from a small to large failure events for damage (Capelli

et al., 2018) that may lead to a brittle failure within snow. Also, efforts were made in past to relate the snow stability to the low
frequency acoustic signatures or seismic waves generated from snow preceding the avalanches (Gubler, 1979; Sommerfeld and
Gubler, 1983; Sommerfeld, 1982; van Herwijnen and Schweizer, 2011); however, linking these signals to the snow stability has





still been elusive (van Herwijnen and Schweizer, 2011). To the best of our knowledge, no such work is reported till this date, where AE has been applied to monitor the avalanche formation and release processes confirmed by natural avalanche event.

The motivation behind present work is to explore the capability of AE technique towards non-invasive and passive detection of the AE signatures produced by snow during avalanche formation and release processes. The primary aim of present work is to register the AE activity from a snowpack built over the avalanche starting zone through use of AE sensor-arrestor arrays and high speed multichannel AE acquisition system. Secondly, it is aimed to analyze and correlate the AE parameters detected from the snowpack to the instability development processes prior to the avalanche release. For quantitative assessment of the

snowpack instability, a method is formulated for computation of AE based instability index which is applied on the analysis of voluminous AE data collected during avalanche formation event. A window-wise analysis is presented for AE detected by different sensors mounted on avalanche starting zone. Further, an attempt is made to link the instability index to the avalanche initiation and release processes, once the critical instability was triggered within the snowpack. Different states of the snowpack instabilities are assigned corresponding to the transitions across different states of the snowpack.

## 2   Methods and Data Acquisition

### 2.1   AE-derived Instability index ($\beta_{in}$)

Acoustic signatures are released during several physical processes (events) such as failure, damage, bond-cleavage, dislocation movement, deformation, metamorphism, rupture, phase transition, etc. covering a wide frequency range. The AE released during several physical processes including the melt-freeze metamorphism of a snowpack under natural environment is reported

by Kapil et al. (2014). An AE sensor detects these acoustic (AE) waves in the form of analogue voltage signals (electrical waveform) termed as a hit (Fig.1). The duration of a hit may vary from micro-seconds to multiple seconds depending upon the event size. In the domain of AE, a hit is actually a representative of an event that can be detected by a sensor. In a snowpack, the magnitude of AE signal depends primarily on the types of the event (physical process) behind AE release, i.e. bond-failure, brittle failures, cracking, rupture, etc., which may be associated to larger magnitudes of AE as compared to the processes like

ductile failure, creep, metamorphism, densification, wind drift, etc. The selection of threshold level (in volts or in dB) is essential to suppress the undesired signals and noise components; in Fig.1, the threshold level is represented by a horizontal dark red line. Several AE parameters can be extracted out of a hit (Mistras Group Inc., 2011). The numbers of voltage fluctuations (ripples) above threshold level are termed as count which are shown by black dashed curves ( Fig. 1). The AE-amplitude is defined as voltage signal corresponding to maximum (positive or negative) value of an un-amplified AE signal ($V_{max}$) and

can be expressed in dB. Also, AE energy is defined by integral of the mean squared voltage over the hit duration divided by a reference resistance and is expressed in terms of Joule or atto Joule ($10^{-18}$ Joule). Further, the AE signal strength can be expressed by integral of rectified voltage signal over hit duration and is measured in the units of $10^{-12}$ Vs. The average signal level (ASL) is a measure of averaged amplitude of the signal expressed in dB.

The mechanism of snowpack instability evolution within a snowpack and further triggering (transition) of instability, from

equilibrium (stable) to unstable state, is quite essential to study the avalanche formation and release processes. The instability





within a weak layer of a snowpack may be triggered because of internal or external impulses and thereafter the progressive increase in the instability may result into the slab failure. Using AE technique, it may be possible to detect the acoustic signatures produced during the ongoing failure activities which may start from the microscopic (crystal or grain level) to macroscopic (crack propagation or slab failure) scales. For easier and quantitative interpretation of AE released during instability development processes, a method is proposed to assess the levels of instability prevailing within a snowpack directly in terms of instability index. In this approach, the instability index is derived from two independent and uncorrelated AE variables of orthogonal nature (expressed along two mutually perpendicular axes) such as amplitude and count (Fig. 1). Collectively, these two variables (count and amplitude) can represent the effective extent of an AE signal. The number of counts may better resolve the information as compared to the number of hits detected during an AE activity; as the counts are derived from the hit itself.

The center of gravity (c.g.) of any data scatter corresponding to a particular interval of time (window) can be computed from the count $(n_i)$ and amplitude $(x_i)$, following the relation:

$X_{cg} = \frac{\sum_{i=1}^{N} n_i x_i}{\sum_{i=1}^{N} n_i}$ (1)

where '$X'_{TH}$ is the threshold level (in dB). Ideally, no AEs should be released from a stable system, i.e. $X_{cg}(stable) \approx X_{TH}$. The deviation in c.g. of AE data set corresponding to an unstable system ($X_{cg}$: unstable) to that of a stable system ($X_{cg}$ :

$stable \approx X_N$) could be the measure of ongoing instability. For a particular window, the $X_{cg}$ represents the average distribution of the AE amplitude with respect to the counts; therefore, $X_{cg}$ could be a parameter yielding information about the temporal evolution of instability for all running windows. Here, we also define various states of the snowpack instabilities such as stable $X_S$, subcritical $X_{SbC}$, critical $X_C$ and super critical $X_{SC}$. The super critical stability level ($X_{SC}$), assigned with respect to the threshold level $X_{TH}$, lies within the plausible range of instability of a system and is related to $X_{TH}$ by $X_{SC} = \frac{X_{TH}}{log(e)}$. Here,

the term log (e) has arisen while transferring the exponential (power-law) behavior of AE signal (voltage) to the logarithm (dB) scale, following the work of Utsu (1966), and Tinti et al. (1987). In present work, the combined effect of AE amplitude and count are expressed in the form of an index, termed as instability index ($\beta_{in}$). Here, the $\beta_{in}$ is defined as a measure of the deviation of center of gravity (c.g.) of a particular AE data scatter with respect to the c.g. of stable system. The $\beta_{in}$ (a normalized index) is defined by

$\beta_{in} = \frac{X_{cg} - X_{TH}}{X_{SC} - X_{TH}}$ (2)

The abrupt changes in $\beta_{in}$ are referred to the transitions between two different states of the snowpack instabilities which may be taking place between the stable and unstable states. Beyond $X_{SC}$, the system may switch over to a region of catastrophic failure (rupture). The triggering of critical instability within snow can be associated to the large AE events produced particularly during brittle failure activity within snow and represented by sharp jumps in the $\beta_{in}$-values with respect to the past moderate

or low level AE activity. The instability evolving within a system could be correlated to the progressive failure or damage process. Particularly, in a material like snow where bond-failure and bond-formation (sintering) processes occur spontaneously, the trends of $\beta_{in}$ could be thus used as quantitative indicator of the instability evolving during avalanche initiation process.





## 2.2 AE Sensor and Multichannel AE System

A broadband and high speed multichannel AE system (Mistras Group Inc., 2011, PAC, USA) was used to acquire the AE data
from the avalanche slope. Two types of AE sensors (resonant and non-resonant), in a wide frequency range, were deployed
in avalanche starting zone for detection of AE activity from snowpack. The characteristic design and details about all AE
sensors used in present study are presented under Table 1. Although, an array of twelve different AE sensors was installed in
the beginning only six sensors (S1, S2, S4, S5, S7 and S8) could work properly during the study period (16 Feb 2016 to 22
Feb 2016) pertaining to the avalanche formation and release event. During the study period (16 Feb 2016 (11:07 am) to 22 Feb
2016 (7:43 pm)), nearly 56 GB of the AE data was recorded by six different active sensors. The AE data acquisition system
(DAQ) was used to acquire, filter, process and store the AE data where a 16-bit ADC (analogue to digital converter) was used to
digitize the AE analogue signal at sample rate up to 1 MSPS. In this system, the range of AE counts may vary from 0 to 65,535
and the range of AE amplitude can be from 10 to 100 dB (0 dB ref. 1V/m/s). Further, the threshold level can be set above 27
dB for removing the undesired noise components, but below the threshold of 27dB, the instrumental noise of the AE system
interrupt the AE signal. In accordance with the prescribed frequency range, the software selectable (appropriate) options are
also in-built within the DAQ for both the analogue filters (LP, HP) and digital filters (4th order Butterworth). Further to achieve
better SNR, the acoustic impedance matching between sensor and snow is quite essential which helps in detection of feeble
AE activities from a snowpack. The attenuation of AE-waves in snow decreases the signal magnitudes (Faillettaz et al., 2016;
Capelli et al., 2016), therefore, to overcome this problem, specialized AE antenna (arrestors) are required for detection of the
feeble AE. For design of the AE arrestors, past research work of Kapil et al. (2014), Smith and Dixon (2014) and Dixon et al.
(2018) are cited. The AE arrestors of different designs and dimensions were fabricated and then tested for an optimum design in
view of the conditions of a natural avalanche slope. In this work, two different designs of AE arrestors i.e., cylindrical (diameter
- 20 cm, height - 50 cm) and the parabolic (diameter – 40 cm, height - 20 cm, depth - 10 cm) arrestors were fabricated from
aluminum sheets of the thickness 4.0 mm, as shown in Figs. 2 (a, b). A robust housing arrangement (Fig. 2c) is also provided
to protect the AE sensor under harsh environmental conditions, and further to prevent any kind of lateral movements of the
sensors with respect to arrestor surface. For better coupling, high vacuum silicon compound was used as a couplant both for
sensor - arrestor and for arrestor-snow coupling (Kapil et al., 2014); the silicon compound is helpful particularly at sub-zero
temperature conditions pertaining to an avalanche terrain.

## 2.3 Study site and AE Instrumentation on Avalanche slope

A small and accessible avalanche site was identified for erection of AE sensor array which is located near Patsio in the Great
Himalayan range of India (latitude $32^0 45'18''$ N, longitude $77^0 15'36.9''$ E) at an altitude of 3800 m asl. The avalanche starting
zone was demarcated on the basis of the slope angle ($30^0$ to $50^0$) and also by considering the history and frequency of avalanche
occurrence for this site. An aerial view of the study site (image collected during onset of the winter 2016) is shown in Fig. 3(a).
In this figure, the avalanche starting zone is demarcated by a black dashed line, and a fracture line (post avalanche release)
is marked by yellow dashed line. In present case, the average inclination of the avalanche starting zone is approximately $34^0$.





The total covered area of the starting zone is about 618 m², with major and minor axes as 74 m and 32 m, respectively. The topography (surface) of the formation zone is smooth and barren with some portion filled with scree and small boulders. A convex bump near upper section of the slope, as shown in Fig 3(a), may possibly introduce some discontinuity in the snowpack structure where probable failure and fracture arrest (Jamieson and Johnston, 1992) in the snowpack are more likely. For an

optimized sensor array, the idea about probable fracture zone is quite essential that may be identified through past history of snowpack failure, convexity features, continuity in slope angle, simulation studies, etc. The schematic design of AE sensor array on avalanche slope is presented in Fig. 3(b). Further, to find the AE arrestors particularly during the winter season (when arrestors remain buried well into snow), bamboo poles (height 3m, with colored bands) were also erected along the ridge line of the avalanche starting zone. In this figure, the positions of the respective poles are represented by the dark black circles

and the positions of arrestor are shown by lighter circles. In present setup, the arrestors were fixed at the bottom layer of the snowpack (during the onset of the winter season when snow height was about 0.3 m) and were not anchored to the ground. Here, the arrestor-to-arrestor distance was kept nearly 10 m. The details of AE sensor configuration with its relative arrestor positions are mentioned under Table 2.

## 3    Experiments and Data

### 170    3.1    Registration of Avalanche event

The AE sensor array, installed over the demarcated avalanche slope, was made functional on 28 January 2016 to acquire the AE activity continuously. Other supporting data such as snow-met conditions, snowpack layer stratigraphy and stability information were also collected. Any avalanche activity or fracture line visible over snow surface was continuously monitored by team deployed there. From the study site, an avalanche event was registered on 20 Feb 2016 at 11:49 am (IST, Indian Standard Time).

Further, in the morning hours of 19 Feb 2016, snowfall started in this region (around 7:10 am) that continued till 10:20 am on 20 Feb 2016 (IST). A low level AE activity was recorded with the ongoing snowfall in the beginning; however, after 10:41 pm of 19 Feb 2016 (IST), the AE activity increased tremendously at a very high rate, detected by all the active sensors and also in the same fashion. A continuous watch (manual) was kept on the slope thereafter for any probable failure activity, and interestingly an avalanche was observed sharply at 11:49 am on 20 February 2016. Subsequently, other avalanches also triggered in the

adjoining slopes within two hours of the duration. Also, during the previous pit analysis of the snowpack, persistent weak layer were identified near the basal layer of snowpack (discussed in next section) which supported the possibility of an avalanche event. Prior to the onset of snowfall, the AE-arrestors were completely buried into snow. Fig. 4 shows the front view of the avalanche slope, captured immediately after the avalanche event. To discriminate the failed snowpack with respect to white background of snow, a fracture line is also marked over this figure shown by a red dotted line.



## 3.2 Snow-met, stratigraphic and stability survey for avalanche occurrence

In addition to AE data, the other supporting data was also collected from the snowpack on avalanche slope in order to get more insights about the avalanche formation and release processes. The information collected on snow- meteorological conditions, snowpack layer structure, stability and weak layer are presented under Tables 3 and 4 for study period (16 Feb 2016 to 22 Feb 2016). The snowpack data was collected regularly from a nearby observatory of SASE, twice a day (in the morning hours at 08:30 am and in evening hours at 05:30 pm, IST). The observatory is located at a distance of nearly 180 m from the selected avalanche slope. A continuous snowfall took place during the study period from 19 Feb 2016 (07:10 am) to 20 Feb 2016 (10:20 am) which increased the standing snow height from 65 to 104 cm. During the study period, the wind speed varied from 1.1 to 4.8 ms$^{-1}$; however, during the snowfall period, the wind speed increased from 2.7 to 4.8 ms$^{-1}$. The snow surface temperature was found in the range of $-3.2^0C$ to $-22.0^0C$. Also, the snow (fresh) density was found between 150 and 170 kg m$^{-3}$. Out of the total study period of 7 days (16 Feb 2016 to 22 Feb 2016), only 2 days (19 and 20 Feb 2016) were the snowfall days and remaining 5 days were dry.

Further, to get the layer information of the snowpack in addition to the presence of persistent weak layer, the pit analysis (stratigraphy) was conducted on weekly basis from a nearby snowpack. But after avalanche event, it was conducted on the actual avalanche site. The post avalanche releases stratigraphic survey (Table 4) of the snowpack (conducted on 21 Feb 2016) show six major layers of thickness greater than 9 cm which were almost similar to that conducted previously on 16 Feb 2019 except the top most layer of the fresh snow. Particularly in context of the avalanche initiation and release, the confirmation of persistent weak layer within the snowpack was quite essential. In the pit-analysis (Table 4), it was observed that the basal layer (thickness - 23 cm) was composed of depth hoar (DH) crystals of size between 2 to 3 mm. Also, the next two layers above the bottom DH-layer, the faceted crystal (FC) layers of thicknesses 7 cm and 10 cm were also identified. Here, the snow particle classification considered is in accordance with the scheme published by Fierz et al. (2009). For assessment of the snowpack stability, the shovel and column tests were conducted after the avalanche release. These stability tests have also confirmed the presence of weak layers in vicinity of the bottom most layer of the snowpack (up to height of 40 cm). Further, the stability index (SI), measured following the method suggested by Schweizer et al. (2003), were found to be 1.08 and 0.67 prior to avalanche (16 Feb 2016) and post avalanche event (21 Feb 2016) respectively.

## 4 Results

### 4.1 AE parameters pertaining to avalanche formation and release period

Some of the AE parameters registered during the study period pertaining to formation and release processes of a natural avalanche are reported here. The activity corresponding to three prominent AE parameters viz, count, amplitude and signal strength, recorded during the study period (16 Feb 2016 (11:07 am) to 22 Feb 2016 (7:43 pm)) are presented in Figs. 5 (a, b, c). These scatters are actually the replay of the actual AE activity monitored during avalanche event (detected by sensor S1) which is yielding a comprehensive overview of the entire course exhibited in terms of AE (for a period of nearly one week prior to





the avalanche release). These results correspond to the AE sensor S1 as shown in Figs. 5 (a, b, c) which has actually detected the best AE response among other sensors (because it was mounted on the AE arrestor that was closest to the fracture line, Fig. 4). For better understanding of the AE activity, the mean values corresponding to count, amplitude and signal strengths

are also estimated for each running windows of size 10 minutes (600s) and represented by yellow lines over respective plots (Figs. 5 (a, b, c)). In all the figures, the snowfall period (for 19 February to 20 February 2016), is shown by shaded region. In our observations, no major AE activity was there except nearly 13 hours before the avalanche release event (registered on 20 Feb 2016 at 11:49 am, IST). However, a substantial increase in AE activity was also observed in the beginning of the study period (16 Feb 2016) which could possibly be due to direct hitting of snow particles on the exposed topmost portions of the

AE arrestors inside the snowpack (during a light snowfall occurred during early hours of 16 Feb 2016). Thereafter, the AE arrestors were completely buried inside the snowpack. Moreover, in the beginning of snowfall for period 19 Feb 2016 to 20 Feb 2016, the AE activity was observed relatively at a lower rate in spite of the high snowfall rate and high wind activity. But surprisingly, later in the night hours of 19 Feb 2016, an abrupt increase in the AE activity was observed (precisely after 10:41 pm, 19 Feb 2016) that formed a sharp dense 'tower' in terms of the AE counts (as shown in Fig 5a) which remained there for

nearly 13 hours until the avalanche occured (registered precisely at 11:49 am, 20 Feb 2016). Interestingly, a sharp decrease in the AE counts was detected after the release of avalanche. A similar response is demonstrated by all the AE sensors mounted on avalanche formation zone at different locations. Further, with start of the snowfall (for period 19 Feb 2016 to 20 Feb 2016), there was an appreciable increase in the AE amplitude but later on increasing trend was observed in a step-wise fashion (Fig. 5b). Also, the AE signal strengths recorded during above period has revealed a clear and increasing trend (Fig. 5c) having

difference of many orders particularly during the last 13 critical hours prior to the avalanche release event. A large scatter is observed in case of AE amplitude (Fig. 5b) as compared to AE count (Fig. 5a) and signal strength (Fig. 5c). But overall an increasing trend in amplitude is there until the release of avalanche. The larger scatter of the amplitude is due to the logarithmic scale (in dB) where amplitude values are compressed between 0 to 100 dB with respect to a linear scale as used in the case of counts (varying between 0 to 65535 counts) despite of the total numbers of the AE-hits in all the plots to be same. The top

clipping (at 65535) in the case of count plot is due to maximum processing limit of DAQ where count has been taken as a 2-byte value. Further, from Figs. 5 (a, b, c) the onset of dense and increasing trend of AE amplitude seems to be little earlier than the sharp band observed in the case of AE counts as well as in the case of signal strength because of the scale used to display these parameters. Furthermore, the spectral response of each AE hit detected during the study period is interpreted in terms of the frequency centroid (FC) and is shown in Fig 5d corresponding to sensor S1. From our investigations a broadband

FC spectrum, centred about the frequency 60 kHz, was observed particularly during the instability development phase where a dense envelope is formed between the frequency 30 kHz to 90 kHz (Fig. 5d). During the avalanche formation hours the scatter density was very large as compared to other normal conditions. From our experimental investigations for a natural avalanche event, the AE activity was detected at a very high rate, and about a total of 11 million AE-hits were detected by six different active sensors deployed on avalanche slope, during the critical hours prior to the avalanche release event. The authors are of

the opinion that prior to the release of avalanche, the dense tower in terms of AE counts could possibly represent the critical hours of instability development of the avalanche formation period. Nearly 13 hours prior to the avalanche release event can





therefore be attributed to the critical period of the avalanche formation or critical instability development period, as confirmed by the magnitudes and progressive trends of the AE activity prior to the avalanche release. A sharp decrease in the AE activity after the avalanche release event indicates that instability development process has culminated following the avalanche release

and snowpack has stabilized.

## 4.2    Window-wise analysis of relevant AE parameters pertaining to avalanche formation and release processes

The bond fracturing and bond formation are two dominant processes (Schweizer, 1999) spontaneously occurring within a snowpack which may affect the properties and strength of snow, and consequently the instability development process within a snowpack. The bond formation (sintering) and failures in snow (or ice) are observed in the scales of seconds (Szabo and

Schneebeli, 2007; Peinke et al., 2019) to the scales of hours (Colbeck, 1997; Birkeland et al., 2006; Podolskiy et al., 2014). Therefore, the strength (Reiweger et al., 2009) and other physical properties of a snowpack depend on the scales of sintering (healing) and failure processes. It is quite difficult to visualize the ongoing failure and healing processes together. In context of such complexities, the AE analysis is presented with respect to two different window scales, i.e. small window (in the scales of seconds) and large window (in the scales of hours) in order to understand the entire instability evolution process

of a snowpack preceding the avalanche. The large window can represent the averaged out responses of AE activities due to small scale instabilities such as bond fracturing (cracking) and bond formation. For better understanding about the snowpack instability development process, a window-wise analysis may be helpful to interpret the AE information. In this section, some relevant AE parameters, such as hit duration, AE energy and ASL, are estimated for an optimum window size of 10 minutes (w: 600 s) particularly during critical hours of the avalanche formation and the results are shown in Figs. 6 (a, b, c) corresponding

to three different AE sensors S1, S5 and S8. Here, the snowfall period is shown by shaded regions. For comprehensive and better interpretation of the large AE data (56 GB), only three sensors (S1, S5 and S8) are considered because of their respective positions on the slope so as to cover the entire extent of the avalanche formation zone. The hit duration is one of the important information derived from AE and can be correlated to the failure event of a bond detected by the sensor. In present study, the maximum value of the AE hit duration are estimated for window of duration 600 s and the results are presented in Fig. 6(a) with

respect to the sensors S1, S5 and S8. This figure indicates that the hit duration is saturated, particularly for the critical period of avalanche formation, at a level of $10^6$ micro-seconds corresponding to sensors S1 and S8. The hit duration corresponding to sensor S5, except for a while just before the avalanche release, was observed relatively smaller as compared to other sensors S1 and S8. In present DAQ, the maximum limit of the hit duration is 1s; therefore, the saturation in hit duration has occurred after attaining its maximum level. Our observations revealed that during the critical period of avalanche formation, the extreme

AE activity has resulted in the hit duration of longest period.

Further, the AE energy provides a vital information about the mechanical energy released during any failure or cracking activity and can be correlated to the bond-energy or fracture energy of the material. The magnitude of AE energy is an important indicator of the microscopic failure activities during the instability development process. In this work, the maximum value for AE energy are estimated for an optimum window size of 600 seconds and the results are represented in Fig.6 (b) with respect

to the sensors S1, S5 and S8. This plot clearly demonstrates that during the critical hours of avalanche formation, there are





multiple (sharp) peaks of AE energy with increasing heights approaching towards the avalanche release. The increasing peak heights of AE energy thus could be correlated to the development of instability within the snowpack. The average signal level (ASL) could also be used as an indicator for AE activity and here it has been estimated within running window of size 600 s and results are shown in Fig. 6(c). In the case of ASL, an increasing trend is exhibited by all the sensors except by sensor

S5; the overall response of S5 is below the sensors S1 and S8. Also, the ASL peak position for S5 is having a time-lag with respect to the sensors S1 and S8. The time-lag between the peak position detected by sensor S5 and that of the other sensors is about 2786s which could be because of some unnoticed local failures in vicinity of the sensor S5. It could be due to the relative position and shape of the arrestor (Ar3, parabolic) comprising the sensor S5, which was away from the fracture line (Fig.4) as compared to other sensors.

### 4.3 Instability index in relation to avalanche formation and release processes

For better and comprehensive representation of the failure activities occurring within a snowpack, the instability index $(\beta_{in})$ is derived from AE amplitude and counts following the method discussed in section 2.1. In this section, the $\beta_{in}$ is computed, using Eq (2), for all active sensors and corresponding to two different window scales, i.e. smaller and larges sizes. In the larger window, the window size (time interval) varied from 26 minutes to 96 minutes depending up on the variable rates of

AE activity during critical and extreme hours of the avalanche formation and release processes. Here, a larger window size corresponds to the time taken to occupy the memory space of 2 GB (fixed) for raw AE data received at random and variable rates during different phases of the instability development in snowpack. Here, the selection of larger window size may also help in overcoming the fluctuations and scatter in AE data. In case of smaller window, the window size is kept constant (w: 60 s). In present case, the $\beta_{in}$ values are computed over the large window size during the critical hours of avalanche

formation, shown in Fig.7, corresponding to five different AE sensors (S1, S2, S5, S7 and S8) which could detect the AE activity prominently. Here, the time (along x-axis) corresponding to each $\beta_{in}$ value is the mid-point of the window (varying from 26 to 96 minutes), which is slightly deviated with respect to the actual time of any observation and also from actual time of the avalanche release. The snowfall period is shown by the shaded regions. Fig. 7 represents a comprehensive overview of the ongoing instability development process, reflected in terms of the instability index which clearly indicates an increasing

trend of $\beta_{in}$ values starting from triggering time of critical instability until the release of avalanche. The triggering of critical instability is assigned to abrupt jump in the $\beta_{in}$ values prior to the imminent failure of the snowpack, for all the sensors. Further, a linear least square fit is shown for S1 (best responding sensor), drawn over the Fig. 7, between the interval 300331 s and 341874 s (demarcated by black dashed lines). This interval could possibly be in the proximity of the most critical period of the avalanche formation where the progressive damages within the snowpack could be most dominant. In present case,

the maximum value of $\beta_{in}$ is found to be 1.04 for sensor S1. The value of $\beta_{in}$ corresponding to actual failure $(\beta_{CF})$ of the snowpack was extrapolated to 1.14 for large window analyses. Corresponding to small window scale, a minute-wise (w: 60s) assessment of snowpack instabilities are also presented in terms of $\beta_{in}$ corresponding to all active sensors (S1, S2, S4, S5, S7 and S8) for the critical hours of avalanche formation and the results are presented in Figs. 8 (a, b, c, d, e and f). These figures clearly indicate that all active sensors (S1, S2, S4, S7, S8) except S5, have shown an increasing trends of $\beta_{in}$, particularly





during the critical hours of avalanche formation. The least square linear fit for $\beta_{in}$ values are also estimated and depicted over the respective plots for critical period of instability development i.e. time (s) : 300046 s to 348283 s corresponding to all the sensors, except sensor S5. For sensor S5, initially no clear trend was observed but later (between time (s): 331821 s to 349303 s) an increasing trend in $\beta_{in}$ was observed before the avalanche release. Also, the slope of each curve-fit indicates the rate of progressive damage process within the snowpack during avalanche formation. From these observations, the maximum values

of $\beta_{in}$ for the best responding sensors S1, S4 and S7, were estimated to be 1.28, 1.21 and 1.21, respectively.

### 4.4    Instability index ($\beta_{in}$) based approach for classification of snowpack instability states

The instability within a system may arise whenever its equilibrium state is disturbed due to action of some unbalanced forces induced by any internal or external cause. A transition from a stable to an unstable structure (Capelli et al., 2018) may lead to a failure. The snowpack or a slab on avalanche slope may switch over from static to a quasi-static and then to a dynamic system

during the course of avalanche formation and release processes. A stable snowpack can be a static system but on other hand, an unstable snowpack may pass through different quasi-static states which may give rise to different states of the snowpack instabilities. An avalanche process is a state of dynamic instability (chaos). In this work, an attempt is made to interpret the AE based information for entire duration of the avalanche formation towards identification of different states of instabilities associated to a snowpack on avalanche slope. It is clear from previous section (Figs. 7 and 8) that after triggering of critical

instability within the snowpack, which gave rise to the avalanche formation process (13 hours prior to avalanche release), there is a sharp increase in the $\beta_{in}$ values at certain instances which is demonstrated by all the sensors almost in a similar fashion. The snowpack states are attributed to transition across two successive instability states through abrupt changes in $\beta_{in}$ values. The states of the snowpack(conceptual) are marked over the Figs. 9 (a, b) where sharp jumps in $\beta_{in}$ values are marked both for larger as well as smaller window scales. The transition points for different successive states of the snowpack are shown by

labels A, B, C and D. Fig. 9(a) represents the instability index ($\beta_{in}$) corresponding to large window (lw) for entire duration of the study period (16 Feb 2016 to 22 Feb 2016), and further, for smaller window size (sw), the instability states in terms of $\beta_{in}$ are displayed in Fig. 9(b). The transition associated with different states of snowpack instabilities is clearly marked over the respective figures at levels (A, B, C, D) for the best responding sensor S1. These points are possibly attributed to the transition between different successive states (levels) of the snowpack leading towards a global failure. Here, different

instability states of the snowpack (conceptual) are defined in terms of AE derived instability index ($\beta_{in}$) and are presented under Table 5. The dotted horizontal lines, drawn over the Figs. 9(a, b), are demarcating the boundary of a particular state associated to certain threshold value of $\beta_{in}$. In our assessment, four different transitions, with respect to points A, B, C, D, are assigned corresponding to five different states of the snowpack (Table 4). A stable state is considered for which the level of instability index is considered below 0.38 for both the window sizes i.e. $\beta_{in}$ (lw) < 0.38 and also $\beta_{in}$ (sw) < 0.38. It is

assumed that a transition $A \rightarrow B$, might have occurred between a stable and unstable (sub-critically unstable) state of the snowpack. A sub-critical unstable state is considered where $\beta_{in}$ may vary between 0.38 to 0.65 for large window and between 0.38 to 0.73 for smaller window scales. A critically unstable state is defined for which the transition is $B \rightarrow C$. In case of large window analyses, a critical state is defined for which $\beta_{in}$ may increase from 0.65 to 1.04 and for small window scale,





it may vary from 0.73 to 1.28. A super critically unstable state is defined corresponding to the transition $C \rightarrow D$, where the
instability index may switch over from 1.04 to 1.14 for large window analyses and from 1.28 to 1.36 for small window. The
catastrophic failure of a snowpack or slab is the state of global failure just before the avalanche release time. In present case,
the instability index for the region of catastrophic failure of a snowpack is also determined after extrapolating the linear fit
for $\beta_{in}$ plots, and assigned to $\beta_{in}>1.14$ ($\beta_{CF}$) for large window scale and $\beta_{in}> 1.36$ ($\beta_{CF}$) for small window scales. The AE
based $\beta_{in}$ values corresponding to the transition points between different states of the snowpack instabilities are important
indicators inferring the ongoing instability and avalanche formation process inside a snowpack and helpful for non-invasive
and quantitative assessment of the snowpack stability.

## 5 Discussion

### 5.1 Registration of avalanche formation and release event

The avalanche formation process evolves through the course of instability development induced by microscopic failure activi-
ties within a snowpack, and it is pertinent to relate the AE released during these processes towards assessment of the snowpack
instabilities. The process of instability development in snow may start with the failure processes occurring at very microscopic
levels (St Lawrence, 1980) through the process of initiation, nucleation and the propagation of the cracks within a weak layer
(Schweizer et al., 2003; Schweizer et al., 2016) under the influence of a stress field. The stress concentrations may occur at
grain boundary or across inhomogeneity (crack tip) in a weak layer. The strength of an individual bond (intra or inter-granular)
as well as the number of bonds failed during a crack (fracture) development process may contribute towards the fracture energy
released in a failure process and therefore it can be correlated to AE signal. In this study, the critical instability triggered within
snowpack (having buried weak layers), due to its excessive loading during snowfall, is investigated in context of AE along with
other supporting information including snow-meteorological parameters, layer stratigraphy, pit-analysis, weak layer properties
and stability tests. The snow-meteorological data (Table 3) reveals that snowfall in the study site started on 19 Feb 2016 (at
07:10 am) and continued till 20 Feb 2016 (10:20 am) at a low rate (0.5 to 1.5 mm per hour). Further, the pit-analysis and
stratigraphic survey conducted on a nearby snowpack before onset of this snowfall has clearly revealed there were weak layers
of DH-crystals (thickness of 23 cm) near the basal layer accompanied by the faceted crystal layer (thickness 17 cm) above
the DH layer (Table 4). The stability tests (section 3.2) have also confirmed that the snowpack was unstable and the slope
conditions were prone to failure as it was continuously being loaded by new snow. The AE activity was therefore continuously
being recorded where AE rate was quite low in beginning but suddenly after 10:41 pm on 19 Feb 2016, a tremendous increase
in the AE activity was noticed by all the AE sensors mounted on avalanche slope. We also confirmed that there were no major
effects of snowfall and wind activity on this abnormal AE activity. The reason behind the abrupt and extremely high AE activity
could be possibly due to triggering of critical instability within the snowpack due to its excessive loading during the continuous
snowfall.






## 5.2 AE parametric analysis in context of avalanche formation and release

After reaching the critical loading level of the unstable snowpack, the avalanche formation process (instability development) might have triggered that continued and led to the avalanche release. It is assumed that the snowpack instabilities might have started within the weak layer with consistent overloading of the snowpack. In this understanding, the critical instability might
have achieved (around 10:41 pm, 19 Feb 2016) that resulted into very high rate of AE activity (counts) thereafter. After triggering the critical instability, the AE counts have reached the maximum level (65535) with sharp boundaries representing the time of start of avalanche formation and the release of avalanche. In context of the avalanche formation process, the increase in AE amplitude can be correlated to the increased size of the failure event (fracture or bond-failure) and a progressively increasing trend in AE amplitude and signal strengths were observed until the avalanche released which decreased thereafter.

Appreciable differences in the responses for particular arrestor designs were also observed; the performances of the cylindrical arrestors (connected to sensor S1, S2, S4, S7 S8) were found to be better than the parabolic arrestor (connected to sensor S5). The reason behind the poor response of parabolic arrestor was because of its smaller capture area, which is a critical design factor for AE arrestor (Kapil et al., 2014). Also, the surfaces of parabolic arrestor were in the contact of snow from both the sides whereas the cylindrical arrestor was empty from inside. Taking into account the complexity of AE-wave propagation in
snow, it is essential to consider the sensor proximity to the source of AE (line of arrest of cracks, vertical fractures line) as the signal magnitudes may decrease due to the attenuation offered by snow for different sensors. Since sensor S1 was closest to the fracture line; therefore, the AE activity is best detected by sensor S1 amongst all the sensors. In our opinion, the crack initiation and propagation might have started within the weak layer much earlier than the formation of the fracture line visible on snow surface (Fig. 4), but the slab fragmentation process might not have strictly progressed in one direction rather in a piece-wise
manner. In our observation, the role of snowfall on AE activity is not much significant which is clear from Figs. 5 (a, b, c) and also from 6 (a, b, c) that the increased rate of AE activity was observed much later than the start of snowfall. The prominent AE parameters such as counts, amplitude, signal strength, hit duration, AE energy and ASL have clearly represented the avalanche formation and release processes. The increase in AE counts (Fig. 5a) and hit duration (Fig 6a) is closely matching with the time of occurrence of avalanche release but other parameters have shown an increasing trend with their maximum values close to
the time of avalanche release. The progressively increasing peaks of AE energy and ASL (Figs. 6(b, c)) indicate the increased rate of instabilities tending towards the failure of the snowpack.The spectral behaviour of the snowpack is one vital information on the avalanche formation and release. Our observations have revealed a broadband spectrum below 90 kHz of the frequency which could possibly because of the failure phenomenon occurring at two different scales; the snow bond failures occurring at microscopic scales which may yield relatively higher AE frequencies, and the crack extension (at macroscopic scales) may re-
lease relatively the lower frequency AE. The combined effect of AE wave generation and subsequent propagation in the porous medium (snow) may form a complex broadband AE spectrum. This information may sometime further help to understand the avalanche formation and release processes.





## 5.3 Assessment of snowpack instability

Two competing processes such as bond fracturing and bond formation (sintering) are believed to occur within a snowpack preceding the avalanche release (Schweizer, 1999). The characteristic time of failures and sintering processes occurring within a snowpack may affect the snow strength significantly (Reiweger et al., 2009) and consequently, the mechanism of avalanche formation and release. The scales of microscopic (localized cracking in weak layer) and macroscopic (slab fracture) cannot necessarily be same which is due to the complex nature of failure (rupture, bond cleavage) and the sintering (healing) processes

in a snowpack which make the failure prediction of snow rather complicated as compared to other materials like soil, sand etc. In snow, the scales of sintering (healing) may vary from 10 to 1000 milliseconds (Szabo and Schneebeli, 2007; Peinke et al., 2019) to the scales of the hours (Colbeck, 1997; Birkeland et al., 2006; Podolskiy et al., 2014). Therefore, the selection of two different time windows, i.e. small window (in the scales of seconds) and large window (in the scales of hours) are aimed to visualize the entire instability evolution process. The large scale window also allows averaging out the small scale discontinuities

due to bond fracturing and sintering processes. Therefore, a window-wise analysis is presented in terms of few relevant AE parameters for different sensors mounted at different locations on avalanche slope. Following the increasing trend of $\beta_{in}$ corresponding to small as well as for large window scales can be considered analogues to the ongoing (progressive) damage process during instability development phase. The $\beta_{in}$ estimated for all the active sensors have shown increasing trend (almost a linear fashion), once the instability has triggered within the snowpack. The maximum value of $\beta_{in}$, shown by different sensors was

observed closer to the avalanche release time. Also, the end of the snow fall and the maxima of these parameters are not exactly the same. The exact timing of snowfall is not clearly visible in the figure due to long period of the AE activity (resolution of sub-second) compressed in small scale of display. The $\beta_{in}$ values are correlated to the ongoing instabilities within snow which may be in the scales of microscopic (crack initiation and nucleation) to macroscopic (crack propagation) failures. Also, there could be a progressive link between increasing or decreasing trends of $\beta_{in}$ with failure or sintering processes. Contrary to

failure, a sintering (healing) process precludes the increasing trend of $\beta_{in}$. An abrupt jump may correspond to transition (triggering) of the instability and a consistent increase can be correlated to the progressive damage process within the snowpack approaching towards a catastrophic failure.

## 5.4 Conceptualization of snowpack instability states

The entire process of avalanche formation and release involves the switching over of a snowpack from a static (stable) to a quasi-static (critically unstable), and then to a dynamic system (disordered state). In a snowpack, failure (bond-failure) and healing (sintering) processes are occurring spontaneously; therefore several physical states are likely. The avalanche formation and release process cannot simply be defined by two states, i.e. stable and unstable states and there may be several possible unstable states prior to the avalanche release. In our understanding, five different states are conceptualized based on the $\beta_{in}$

analysis inferred from increase in $\beta_{in}$ values following transitions across different states. In ideal conditions, a stable state is defined where no AE is produced but in practice many other physical processes may be generating AE of low magnitudes other





than fracturing process. The critical instability is the level where information is quite significant to understand the progressive damage during critical hours of the avalanche formation. A critically unstable level is achieved through the transition $B \rightarrow C$, which may be followed by $C \rightarrow D$ to attain a super critically unstable state. The threshold values of $\beta_{in}$ are very important alerts (flags), say at level 1.04 large window size and at level 1.28 for small window scale. In case, if $\beta_{in} > 1.0$, there is a possible threat of avalanche release. The catastrophic failure state of any structure may be highly sensitive to the internal or external changes within a transitional state where an unstable system may suddenly switchover to a state of the dynamic instability (chaos) which is rather difficult to predict. The AE would be a helpful tool in predicting even the dynamics of the snowpack state by extrapolating the curve-fit where the AE activity may be observed with very large magnitudes in a non-linear fashion. A consistent increasing trend in $\beta_{in}$ could be a robust indicator for snowpack instability development. Once the critical instability state is triggered, a flag can be assigned in terms of $\beta_{in}$ which may forewarn the upcoming catastrophic failure. Similarly, a flag or precursor may be assigned after triggering of the super critical instability to forewarn the failure of the snowpack. These flags or precursors may be very helpful (key indicators) to predict the failure of a slab or release of an avalanche. Also, the duration of critical instability period, derived from time of triggering of critical instability and the time of occurrence of catastrophic failure state, could be useful information to estimate the avalanche formation period and to predict the avalanche release.

## 6 Conclusions

The objective behind the present study was to acquire, analyze and interpret the AE detected from a snowpack through a case study of the avalanche release event observed in Great Himalaya. AE sensor array (twelve different AE sensors – arrestors) was installed over the avalanche starting zone through a multichannel AE acquisition system. For optimizing the number of sensors over the slope, the most probable fracture zones were identified through past history of snowpack fracture line, convexity features like bumps, continuity in slope angle and terrain topography. For better understanding of the avalanche formation and release processes in relation to AE, the surrounding conditions such as snow-meteorological, snowpack layer stratigraphy, weak layer data and snowpack stability information, were also collected during the study period of over one week. The avalanche event was successfully registered which might have triggered due to continuous excessive loading of an unstable snowpack (combining thick weak layers of DH and faceted crystals) during a snowfall spell. The crack initiation and propagation within the buried weak layer has possibly triggered the critical instability (avalanche formation process) prior to avalanche release. The AE sensors mounted at different locations on avalanche slope have successfully detected the AE activity for entire duration. During critical hours of the avalanche formation (nearly 13 hours prior to the avalanche release) extremely high rate of AE activity was observed. Further, a new method is proposed for easier and faster analysis of snowpack instability derived from volumes of the AE data detected during avalanching process, and same has been applied to compute the snowpack instability index ($\beta_{in}$). The randomness and fluctuations in the AE scatter plot could be mostly due to the combined effect of failure and healing (sintering) processes occurring within a snowpack, spontaneously. A window-based analysis for $\beta_{in}$ is therefore presented to interpret the voluminous AE data. The $\beta_{in}$ values, estimated for all active sensors, have shown an increasing trend


which is correlated to the progressive failure through development of instability, from microscopic to macroscopic scales, after triggering of critical state of the instability and prior to avalanche release. A consistent increase in $\beta_{in}$ values correlated to progressive damage of the snowpack and also an abrupt increase in $\beta_{in}$ values can be attributed to the transitions across different instability states of the snowpack. Four different transitions across five different (conceptual) states of the snowpack are assigned following the abrupt increase in $\beta_{in}$ values, particularly during the period prior to the avalanche release event. The

information retrieved from this study is based on an avalanche event from a specific slope; furthermore studies may help to increase the accuracy of assessment of snowpack instability using this technique. This work could be a step forward to establish AE based non-invasive monitoring of avalanche slope instability in real time and may find many other applications in the areas like landslide, rock fall, structural health monitoring, earthquake etc.

*Author contributions.*  All the authors have contributed significantly in this research paper. JC Kapil developed the methodology for compu-

tation of AE based instability index and technique for AE acquisition from avalanche starting zone. In addition, he contributed in AE data interpretation, discussions, structuring and preparation of this paper. Sakshi carried out the window wise AE data analyses for prominent AE parameters, instability indices through development of MATLAB codes along with the preparation of the first draft of manuscript and graphs, tables, etc. Karmjit Singh was involved in the deployment of AE sensor array on avalanche starting zone, AE data collection along with the collection of snow–met data, pit-analysis, stability tests and slope images. JS Shahi contributed in AE signal analysis and structure

of the paper. Rama Arora contributed in the material selection for design of AE arrestor, language and presentation of this paper.

*Competing interests.*  The authors declare that they have no conflict of interest.

*Acknowledgements.*  Authors express their sincere thanks to Director SASE for his administrative as well as financial support to carry out the research work. We are also thankful to P K Satyawali, P K Srivastava and Prem Datt for extending their administrative supports. Authors acknowledge the support of V. Bharti, M. Midya, K. Kumar and other staff of Cold lab and workshops of SASE during fabrication and

installation of AE sensors array. Also, authors acknowledge the contribution of team involved in the collection of snow-met data, snowpack pit-analysis, stability tests and avalanche related information pertaining to the study site.

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





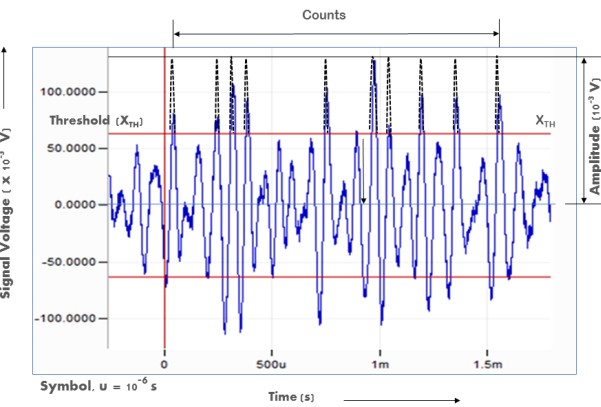

**Figure 1.** A typical waveform representing the extents of AE counts and amplitude, derived from a voltage signal of an AE-wave (hit). In this figure the black dotted curves (fluctuations) are representing the AE counts (the numbers of ripples above threshold level)




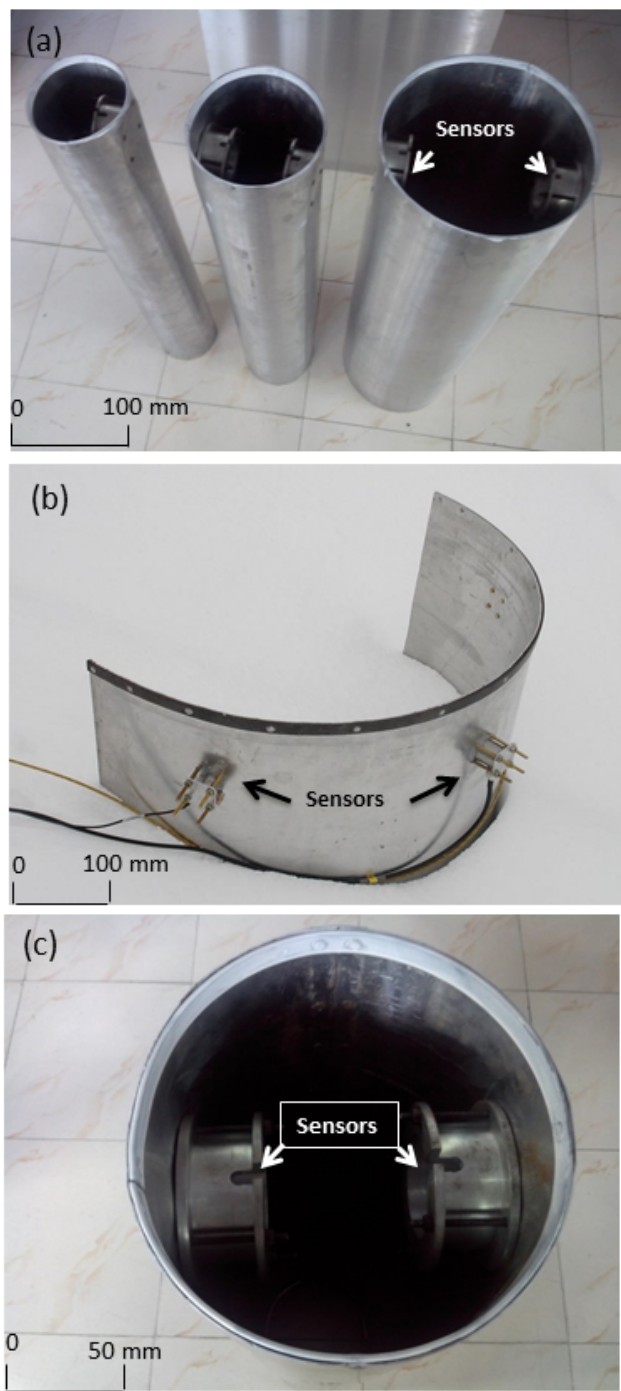

**Figure 2.** AE arrestors (antenna) deployed in the avalanche starting zone for monitoring of avalanche formation and release, (a) Cylindrical AE-Arrestor (b) Parabolic AE-Arrestor (c) Housing arrangement for AE sensor. For dimension see the text.



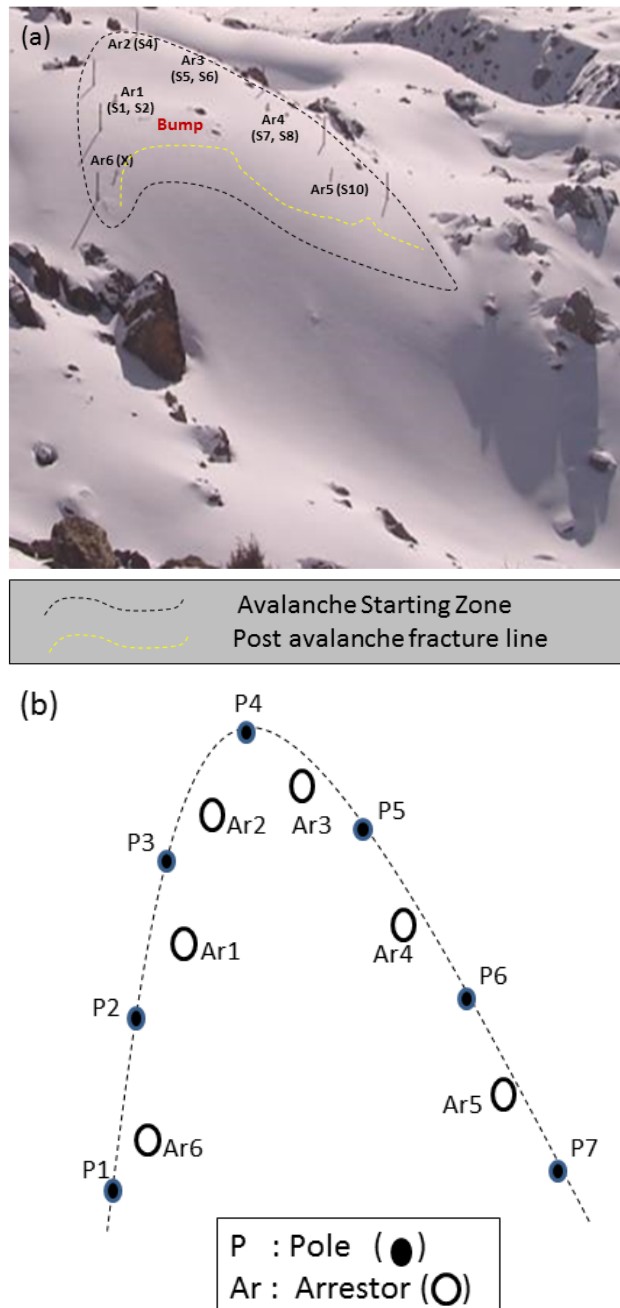

**Figure 3.** (a) Front view of the site showing AE sensor array established on avalanche starting zone near Patsio (altitude 3800 m asl), in Great Himalaya (b) Schematic design for AE sensor array showing the positions of AE arrestors (coupled to sensors S1, S2, S4, S5, S6, S7, S8 and S10) and bamboo poles on avalanche slope. The yellow dotted line represents the fracture line on the snowpack post avalanche release.



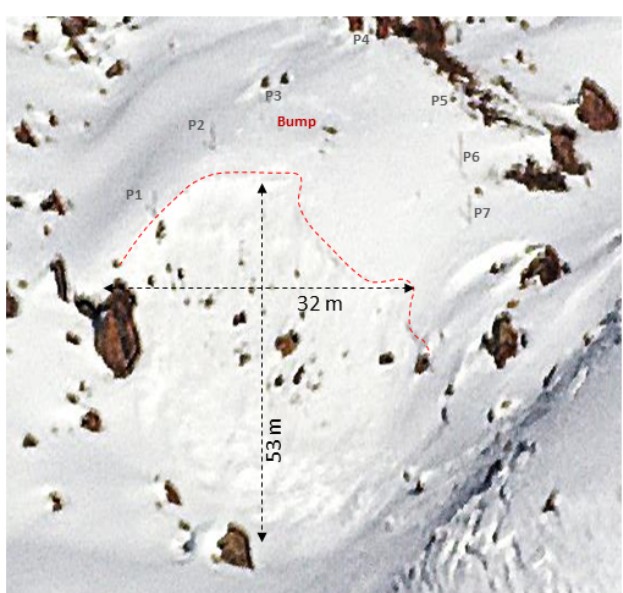

**Figure 4.** Front view of the avalanche released on the selected avalanche site near Patsio (The red dotted line represents the fracture line on the snowpack post avalanche release).



**Figure 5.** Prominent AE characteristics registered during avalanche formation and release processes for sensor S1, (a) AE-Counts, (b) AE-Amplitude, (c) Signal Strength, (d) Frequency Centroid (the start AE acquisition time is 16 Feb 2016 at 11:07 am and the end of AE acquisition time is 21 February 2016 at 07:43 pm). The yellow line in respective figure represents the mean values of the recorded AE parameters estimated over running window of interval 600 s. The two different scales for time (along x-axis) are shown for experimental time in terms of running seconds and also with respect to the local time in IST (Indian Standard Time) and both the scales are synchronized. In all the plots, the shaded region represents the snowfall period (start of snowfall, 19 Feb 2016 at 07:10 am and end of snowfall, 20 Feb 2016 at 10:20 am) and dotted line indicates the time of avalanche occurrence (20 February 2016 at 11:49 am).


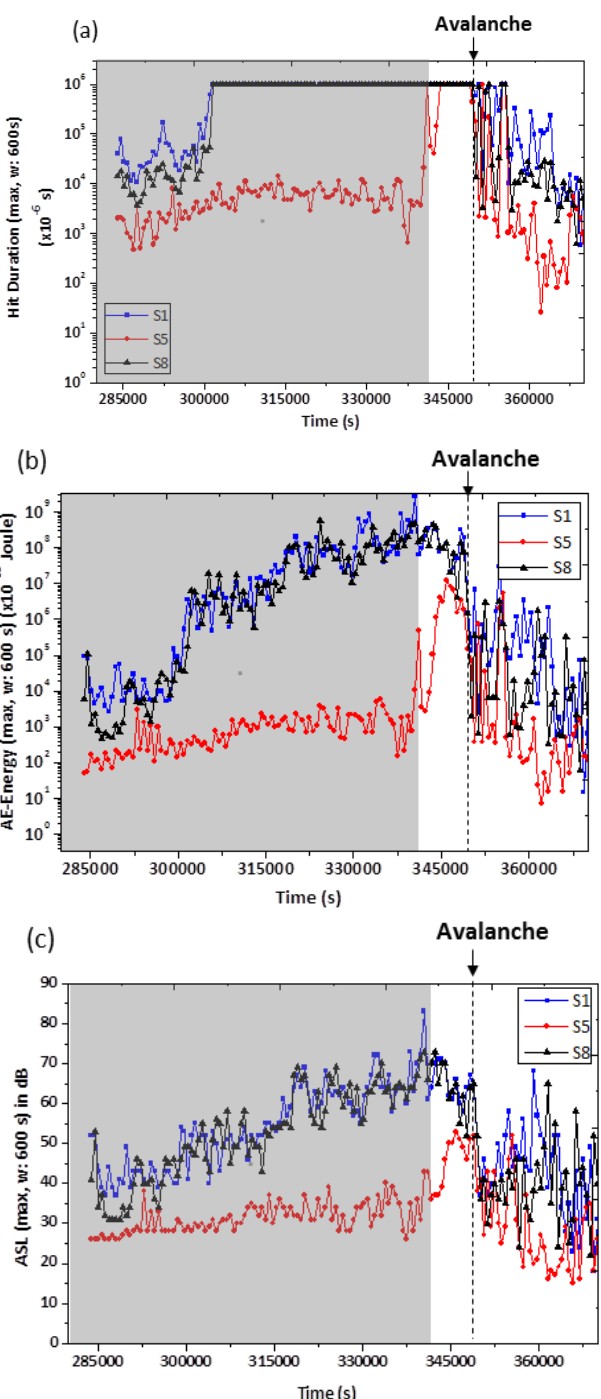

**Figure 6.** Window-wise (window size: 600 s) variation in some prominent AE characteristics captured by AE-Sensors S1, S5 and S8 (placed in slope at different locations) during avalanche formation process, (a) Max-value of Hit Duration, (b) Sum-value of AE-Energy, (c) Max-value of ASL. The dotted line indicates avalanche occurrence time (20 February 2016 at 11:49 am). The saturation observed in the case of AE hit duration (Fig. 6(a)) is due to the limitation of AE acquisition setup after attaining the maximum level of hit duration (1 s).



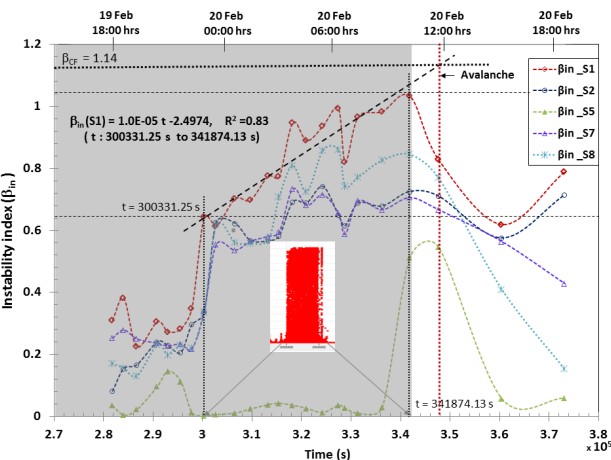

**Figure 7.** Variation in instability index ($\beta_{in}$), estimated for sensors (S1, S2, S5, S7 and S8) detected better the AE activity, during the critical period of avalanche formation and release processes corresponding to the larger window of variable size (w: 26 minutes to 96 minutes) depending upon the AE data rate to occupy constant memory space equivalent to 2GB. The shaded region indicates the snowfall period and the dotted line the time of avalanche release.





**Figure 8.** Variation in instability index ($\beta$in) corresponding to a smaller window scale (w: 60 s) corresponding to different AE Sensors placed on slope, (a) Sensor S1, (b) Sensor S2, (c) Sensor S4 , (d) Sensor S5, (e) Sensor S7, (f) Sensor S8.



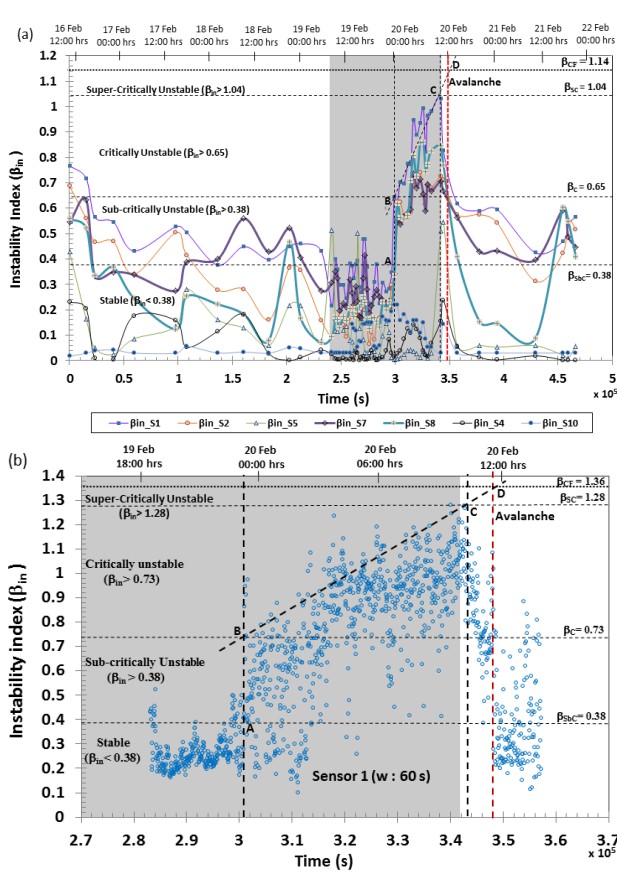

**Figure 9.** Identification of critical transition states for snowpack instability during avalanche formation by monitoring the temporal variation of $\beta_{in}$ corresponding to (a) large (variable) window sizes (w: 26 minutes to 96 minutes), (b) smaller window of fixed interval (w: 60 s) for critical period of avalanche formation. The shaded region indicate the snowfall period and the dotted line the time of avalanche release.





**Table 1.** The details about AE sensor design and characteristics, as used in acquisition of avalanche formation and release processes from a snowpack on avalanche slope. (Physical Acoustics Corporation, USA).

| AE Sensor model[1] | Operating frequency range | Resonant frequency | Detection sensitivity[2] | Operating temperature range |
|---|---|---|---|---|
| R3I-AST | 10 kHz - 40 kHz | 25 kHz | 120 dB | $-35^0$C to $75^0$C |
| R6I-AST | 40 kHz - 100kHz | 55 kHz | 117 dB | $-35^0$C to $75^0$C |
| R3$\alpha$ | 25 kHz - 70 kHz | 29 kHz | 80 dB | $-65^0$C to $175^0$C |
| R15I-AST | 50 kHz- 200 kHz | 75 kHz | 109 dB | $-35^0$C to $75^0$C |
| WDI-AST | 200 kHz -0.9MHz | 125 kHz | 96 dB | $-35^0$C to $75^0$C |
| MICRO 200HF | 500 kHz -4.5 MHz | 2500 kHz | 62 dB | $-65^0$C to $177^0$C |

1: Mistras Group, PAC

2 : 0 dB ref. 1 V.s.m$^{-1}$





**Table 2.** AE sensor configuration and positions with respect to different AE arrestors.

| Arrestor /Type | AE Sensor | AE Sensor model | Acquisition Channel-DAQ | Arrestor position with respect to poles |
|---|---|---|---|---|
| Ar1 Cylindrical | S1 | R3I-AST | Ch1 | P2-Ar1: 1.0 m |
| | S2 | R6I-AST | Ch2 | P3-Ar1:9.5 m |
| Ar2 Cylindrical | S3 | R3$\alpha$ | Ch3 | P3-Ar2: 11.7 m |
| | S4Z | R6I-AST | Ch4 | P4-Ar2:3.0 m |
| Ar3 Parabolic | S5 | R15I-AST | Ch5 | P4-Ar3:3.0 m |
| | S6 | R6I-AST | Ch6 | P5-Ar3:11.8 m |
| Ar4 Cylindrical | S7 | Micro 200HF | Ch7 | P5-Ar4: 1.0 m |
| | S8 | R6I-AST | Ch8 | P6-Ar4: 8.8 m |
| Ar5 Cylindrical | S9 | WDI-AST | Ch9 | P6-Ar5: 8.2 m |
| | S10 | R6I-AST | Ch10 | P7-Ar5: 4.0 m |
| Ar6 Parabolic | S11 | R3I-AST | Ch11 | P1-Ar6: 1.5 m |
| | S12 | R6I-AST | Ch12 | |



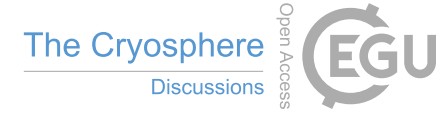

**Table 3.** Snow and meteorological data for study period (16 Feb 2016 to 22 Feb 2016) pertaining to the selected avalanche site.

| Date | Time[1] | Air Temp[2] | Snow surface Temp[3] | Wind speed[4] | Fresh snow amount[5] | New snow density[6] | Snow height[7] | Air Pressure[8] |
|---|---|---|---|---|---|---|---|---|
| 15 Feb-16 | F | -13.0 | -16 | 1.7 | 3 | 80 | 0.66 | 646 |
| | A | -1.5 | -11.5 | 2.1 | 0 | - | 0.65 | 645 |
| 16 Feb-16 | F | -12.5 | -18 | 2.3 | 0 | - | 0.65 | 647 |
| | A | +0.1 | -9 | 1.6 | 0 | - | 0.65 | 647 |
| 17 Feb-16 | F | -16.0 | -22 | 1.5 | 0 | - | 0.65 | 651 |
| | A | +0.5 | -10 | 1.9 | 0 | - | 0.65 | 649 |
| 18 Feb-16 | F | -14.0 | -18.5 | 1.1 | 0 | - | 0.65 | 651 |
| | A | +2.5 | -9 | 2.2 | 0 | - | 0.65 | 650 |
| 19 Feb-16 | F | -8.5 | -6.5 | 0.5 | 0 | - | 0.65 | 651 |
| | A | +0.5 | -6.5 | 2.7 | 18 | 140 | 0.83 | 650 |
| 20 Feb-16 | F | -4.5 | -2.5 | 3.6 | 23 | 150 | 0.106 | 643 |
| | A | +0.5 | -6.5 | 4.8 | 3 | 170 | 0.105 | 646 |
| 21 Feb-16 | F | -15.5 | -21 | 1.1 | 0 | - | 0.104 | 649 |
| | A | +3.5 | -11.5 | 1.8 | 0 | - | 0.100 | 647 |
| 22 Feb-16 | F | -13.0 | -17.5 | 1.0 | 0 | - | 0.100 | 647 |
| | A | +4.0 | -12 | 1.7 | 0 | - | 0.98 | 646 |

1 : F : Fore Noon (measurement time 8:30 am, IST) and A : After Noon (measurement time 05:30 pm, IST).
2,3 : °C
4 : m/s
5 : mm
6: kg/m³
7 : m
8: mb





**Table 4.** Pit-analysis and stratigraphic survey of the snowpack collected post avalanche release (21 Feb 2016) from a snowpack adjacent to the demarcated avalanche site.

| Layer No. | Position[1] | Thickness[2] | Wetness | Grain type[3] | Grain size[4] | Hardness[5] | Density[6] | Shear strength[7] | Water eq.[8] | Snow Temp[9] |
|---|---|---|---|---|---|---|---|---|---|---|
| 1 | 0-23 | 23 | d | DH | 2-3 mm | F | 0.17 | 0.00 | 39.1 | -3.8 |
| 2 | 23-30 | 7 | d | FC | 1-2 mm | F | 0.17 | 5.00 | 11.9 | -3.2 |
| 3 | 30-40 | 10 | d | FC | 1-2 mm | P | 0.24 | 40.00 | 24 | -3.6 |
| 4 | 40-47 | 7 | d | RG | 1 mm | K | 0.25 | 100.00 | 17.5 | -3.9 |
| 5 | 47-57 | 10 | d | RG | 1 mm | K | 0.26 | 90.00 | 26 | -4.6 |
| 6 | 57-64 | 7 | d | DP | <1mm | K | 0.25 | 25.00 | 17.5 | -5.0 |
| 7 | 64-68 | 4 | d | MF | <1mm | 1F | 0.20 | 110.00 | 8 | -5.8 |
| 8 | 68-70 | 8 | d | DP | <1mm | P | 0.22 | 45.00 | 17.6 | -6.6 |
| 9 | 70-79 | 9 | d | DP | 0.5mm | 4F | 0.16 | 58.00 | 14.4 | -7.8 |
| 10 | 79-85 | 6 | d | DP | 1mm | 1F | 0.18 | 33.00 | 10.8 | -12.6 |
| 11 | 85-94 | 9 | d | PP | <1mm | 4F | 0.17 | 25.00 | 15.3 | -20.0 |
| 12 | 94-105 | 11 | d | PP | >1mm | F | 0.12 | 5.00 | 13.2 | -22.0 |

1,2 : cm

3 : Here the acronyms stand for - DH: Depth Hoar, FC: Faceted Crystal, MF: Melt-Freeze, RG: Round Grain, DP: Decomposed Particles, PP: Precipitation Particles (Ref. International Snow Classification Scheme reported by Fierz, et al., 2009).

4 : mm

5 : Hardness Level F: Fist, 1F: 1 Finger, 4F: 4 Fingers, P: Pencil, K: Knife)

6 : g/cc

7 : $Nm^{-2}$

8: Water equivalent in mm

9 : $^\circ$C





**Table 5.** Different states of the snowpack deduced through AE derived instability index (Fig. 9) for avalanche formation and release period corresponding to two different scales of window.

| State of the snowpack | $\beta_{in}$ (lw) | $\beta_{in}$ (sw) |
|---|---|---|
| Stable | $\beta_{in} < 0.38$ ($\beta_S$) | $\beta_{in} < 0.38$ ($\beta_S$) |
| Sub-critically Unstable | $\beta_{in} > 0.38$ ($\beta_{SbC}$) | $\beta_{in} > 0.38$ ($\beta_{SbC}$) |
| Critically Unstable | $\beta_{in} > 0.65$ ($\beta_C$) | $\beta_{in} > 0.73$ ($\beta_C$) |
| Super-Critically Unstable | $\beta_{in} > 1.04$ ($\beta_{SC}$) | $\beta_{in} > 1.28$ ($\beta_{SC}$) |
| Catastrophic Failure | $\beta_{in} > 1.14$ ($\beta_{CF}$) | $\beta_{in} > 1.36$ ($\beta_{CF}$) |

The size of larger window (lw) is variable (varying from 26 to 96 minutes depending upon
the AE data rate to occupy memory space equivalent to 2GB) and the size of smaller window
(sw) is fixed (w: 60 s).