# Peer review of "Acoustic Emission investigation for avalanche formation and release: A case study of dry-slab avalanche event in Great Himalaya"

_The Cryosphere, 2020_

## Referee Comment (RC1) · Anonymous Referee #1 · 11 Mar 2020

The main goal of measuring acoustic emissions (AE) in snow is to predict avalanche release. This paper presents data from AE measurements from an array of sensors placed in an avalanche starting zone prior to the release of a natural avalanche. To the best of my knowledge, these are some of the very first AE measurements prior to catastrophic failure of a snow slope. The data presented in this work are therefore extremely interesting, and the paper therefore warrants publication. However, the paper is currently rather poorly structured, the description of the results is mostly qualitative, the presentation of the data is rather unconvincing and the wording used in the paper is very imprecise. As such, major revisions are required before the paper can be considered for publication.

[Figure]

The manuscript would be much more valuable if the authors focused more of their efforts on predicting / forecasting avalanche release based on AE signal characteristics. Indeed, the authors show there is increased AE activity up to 13 hours prior to avalanche release. However, it remains unclear if this increased activity can be exploited to predict catastrophic failure. In my opinion, a more in-depth analysis in this direction, as for instance in the work of Faillettaz et al. (2015) and the references therein or the work of Capelli et al. (2018), would be the most valuable addition of the paper. In this context, the authors should also consider deriving other parameters from the AE measurements.

The main result of the paper is the definition of five snow instability states based on the observed changes in AE towards avalanche release. The authors arbitrarily define these 5 snow instability states (Table 5) and associate certain threshold values for $\beta$ with these. The definitions of these instability states are not provided. The method used to define the threshold values is also not explained. It thus remains unclear what the differences are between the instability states, and how useful the threshold values are in terms of predicting avalanche release, the stated goal of measuring AE in snow. The authors need to provide a much more convincing case for why five different stability classes are required, and what these mean in terms of failure prediction.

The most simple case would be to have two instability classes (stable and unstable) and evaluate a simple predictive system based on crossing a $\beta$ threshold value, similar to the simple forecasting model suggested by van Herwijnen et al. (2016) (and references therein). Once the $\beta$ value crosses the defined threshold value, an alarm is raised for a defined duration. If an avalanche releases during this alarm, the avalanche was correctly predicted. Such a simple alarm system with two variables ($\beta$ threshold value and alarm duration) could then easily be evaluated in terms of predictive performance (e.g. Probability of detection and False alarm rate for a given threshold value and duration) to obtain the best suited alarm system. Of course, a more complex model could also be envisioned, provided the different instability states are clearly defined.

[Figure]

Overall, the manuscript should contain more quantitative analysis of the data, beyond the rather qualitative description currently given. For instance, the authors suggest that an increase in AE count and signal strength were observed prior to avalanche release. While broadly speaking this is the case, when looking more closely at the figures it becomes clear that prior to the release of the avalanche, both the count and signal strength decreased drastically. This point is not addressed in the manuscript. Perhaps it is related to inaccuracies in the release time of the avalanche. Nevertheless, this should be discussed more thoroughly in the paper. Also, the authors did not analyze signal properties with distance to the avalanche, although this could easily be done with the array of sensors they installed.

Different sensor and arrestor types were used. For me, this complicates the interpretation of the results, as it is not clear to me if the observed differences between the sensors are caused by differences in instrumentation, distance to the avalanche, or related to snow instability? These issues should be addressed more thoroughly in the paper, preferably in a quantified manner. While it is clear to me that this is perhaps difficult to do, at the very least the authors should address this point in the discussion.

The overall structure of the paper is not ideal. There is no clear and succinct description of the site (e.g. Figure 3 should also include a map showing the location of the site), instrumentation and the methods. For instance, the authors mention that the AE sensors were deployed on an avalanche slope. However, it is not specified when and how this was done, and at what depth in the snow cover the sensors were placed. Furthermore, results and discussion are continuously mixed, and the text is often repetitive. The authors should restructure the manuscript and improve the readability. Furthermore, there are many grammatical errors throughout the paper (for instance the first two sentences of the abstract), and I would encourage the authors to seek the help of a native English speaker to go over the manuscript before resubmitting it.

Throughout the paper, there are numerous instances of inaccurate wording, poorly defined terminology and many unfounded statements. For instance, the authors write

that changes in $\beta$ correlated to progressive failure / damage (e.g. lines 437-438; lines 484-485; lines 486-487). However, failure or damage were not measured, and it is thus not possible to correlate both. In this context, it is also not clear how the stability threshold values in Figure 9 were determined. The authors should provide a description of the method/criteria used for this.

The presentation of the data and the results is not very convincing. For instance, the authors should show a figure with the relevant meteorological data (snow height, air temperature and wind speed) of the study period. The authors should also show the snow profile graphically (not a table), show stability test results and the location of the failure layer, and indicate the location of the avalanche failure layer. Figure 4 should show the avalanche that released on the study site. However, I do not see any avalanche debris confirming the release of the avalanche, and the quality of the image is rather poor. Finally, Figure 7 and 8 could be combined. Indeed, for each sensor, the large window average shown in Figure 7 could be plotted on top of the shorter window values shown in Figure 8.

Finally, the authors show data from one snowfall and the subsequent avalanche. It would be very interesting to show data from an additional snowfall without an avalanche, if these data exist. This would provide more insight into the robustness of the observations. Are the results presented in the paper associated with avalanche release, or are the same characteristics also observed for other snowfall events without an avalanche? At the very least, the authors should address this point in the discussion, and more generally the limitations of their results.

More detailed comments can be found in the annotated pdf

Capelli, A., Reiweger, I. and Schweizer, J., 2018. Acoustic emissions signatures prior to snow failure. Journal of Glaciology, 64(246): 543-554. Faillettaz, J., Funk, M. and Vincent, C., 2015. Avalanching glacier instabilities: Review on processes and early warning perspectives. Reviews of Geophysics, 53(2): 203-224. van Herwijnen, A.,

Heck, M. and Schweizer, J., 2016. Forecasting snow avalanches by using avalanche activity data obtained through seismic monitoring. Cold Regions Science and Technology, 132: 68-80.

Please also note the supplement to this comment:
https://www.the-cryosphere-discuss.net/tc-2020-38/tc-2020-38-RC1-supplement.pdf

―――――――――――――――――――――

**Supplement:**

[revised manuscript text omitted]

---

## Referee Comment (RC2) · Anonymous Referee #2 · 14 Apr 2020

I found the paper to be too long for the subject matter. I appreciate the efforts of using AE technique towards avalanche mitigation processes. Its a good tool to have; no doubt. The authors, however, should have refrained from making comments with buzzwords, again and again, on micromechanisms because they seem to be not aware of the "fundamental fact" that snow is an extremely "High-temperature material" undergoing stress-temperature-time induced morphological changes. The micro- and macro-failure mechanisms and their kinetics are very complicated. Nonetheless, significant progresses has been made in the past on physics of failure in polycrystalline solids (not necessarily porous media) at high-homologous temperatures, close to 0.99 Tm, but snow is porous and the authors do not seem to be aware of those developments.

[Figure]

For example, there were AE studies in pure ice for examining the microstructure-property relations - way back in the early 1980's. Specifically, they looked at polycrystalline ice as a "high-temperature" material (existing in our cryosphere at extremely high homologous temperatures, higher than about 0.9 Tm, where Tm is the melting point in Kelvin). Old ideas on AE, based on micromechanisms in other engineering or geological materials at low homologous temperatures should be discarded. However, the use of AE technology can still be used as a tool - such as monitoring the snow states. The authors tried to use AE for real practical application - and should have stayed in that arena - instead of going beyond the realm of the data and speculating about dislocations, etc. They seem to impress the audience with mechanisms that they actually do not fully understand. The manuscript, in this regard, should be modified significantly to bring out the real strength of the work.

---

## Author Comment (AC1) · 8 Jun 2020

**Reply to the comments from Referee 1**

**Acoustic Emission investigation for avalanche formation and release: A case study of dry-slab avalanche event in Great Himalaya (tc-2020-38)**

Below are the point-wise replies (in 'blue' text) from authors against each of the comments made by the reviewer (in 'black' texts) on above manuscript.

**Comment (Ref 1):**

The main goal of measuring acoustic emissions (AE) in snow is to predict avalanche release. This paper presents data from AE measurements from an array of sensors placed in an avalanche starting zone prior to the release of a natural avalanche. To the best of my knowledge, these are some of the very first AE measurements prior to catastrophic failure of a snow slope. The data presented in this work are therefore extremely interesting, and the paper therefore warrants publication. However, the paper is currently rather poorly structured, the description of the results is mostly qualitative, the presentation of the data is rather unconvincing and the wording used in the paper is very imprecise. As such, major revisions are required before the paper can be considered for publication.

**Reply:**

Authors express their thanks to the reviewer for his critical and insightful comments which indeed seems to be very helpful towards improving the quality and presentation of this manuscript. Authors agree to the reviewer's remarks, and in response to each of his suggestions, now the manuscript has been modified and improved significantly.

All the major or minor suggestions/corrections made by the reviewer are incorporated within the revised manuscript. As suggested by the reviewer, to focus on the avalanche prediction using AE, authors agreed upon it, and one sub-section on this aspect is added under the results and discussion sections. A model (appended below), derived from the least square linear fit for $\beta_{in}$, is proposed for avalanche release, and its performance is tested through convergence of the predicted output. As suggested by reviewer, different state of a snowpack are defined through a separate subsection under the method section and a justification is given why two states (stable and unstable) are not sufficient to define the avalanche formation and release processes, completely. The result section has been improved considerably, particularly on the quantitative description of the results. As suggested by reviewer, the AE activity is shown with snowfall rate corresponding to the non-avalanche as well as avalanche period, separately for stable as well as unstable snowpack. The issue of selection of different AE arrestor designs are addressed. A better image of the avalanche release event (showing the debris) is presented here. Also, points regarding location of study site on map, snowpack profile, clubbing of Figs. 7 and 8 are addressed. Further, the snow – meteorological data, pertaining to the study period, are shown graphically.

The manuscript would be much more valuable if the authors focused more of their efforts on predicting / forecasting avalanche release based on AE signal characteristics. Indeed, the authors show there is increased AE activity up to 13 hours prior to avalanche release. However, it remains unclear if this increased activity can be exploited to predict catastrophic failure. In my opinion, a more in-depth analysis in this direction, as for instance in the work of Faillettaz et al. (2015) and the references therein or the work of Capelli et al. (2018), would be the most valuable addition of the paper. In this context, the authors should also consider deriving other parameters from the AE measurements.

Authors agree to the reviewer that an AE based prediction model for avalanche release would certainly be a value addition to this manuscript; however, it requires sufficient AE database and information from multiple avalanche release events. Authors are still in a process to widen the scope of AE based monitoring of avalanches from several avalanche sites of Great Himalayan and Pir-Panjal ranges of Indian Himalaya and development of AE based prediction model. The present work is a case study based on one avalanche event successfully registered in terms of AE.

However, the crucial AE information detected 13 hrs prior to the avalanche release, which has shown nearly a linearly increasing trend, is encouraging, and could be helpful to predict the snowpack failure (avalanche release). Through a linear square fit for $\beta_{in}$, between the onset of critical instability ($t_C$) and the super critical state ($t_{SC}$), a model is formulated for prediction of super critical state. Further extrapolating the same model into the catastrophic regime and also through the introduction of a parameter $\lambda_f$, the model performance is tested with respect to sensor at different locations. The convergence in prediction seems to be increasing while approaching closer towards the release time (actual) of the avalanche. As suggested by the reviewer, the past work of Faillettaz et al. (2015), Capelli et al. (2018), van Herwijnen et al. (2016), etc. are cited and discussed in context of failure prediction. Also, one relevant AE parameter, RMS, is added; however, keeping in mind the length of this manuscript, it will not be possible to add many other AE parameters.

A separate sub-section on AE based prediction for avalanche release is added under the result and discussion, and also the same is appended below.

The main result of the paper is the definition of five snow instability states based on the observed changes in AE towards avalanche release. The authors arbitrarily define these 5 snow instability states (Table 5) and associate certain threshold values for $\beta$ with these. The definitions of these instability states are not provided. The method used to define the threshold values is also not explained. It thus remains unclear what the differences are between the instability states, and how useful the threshold values are in terms of predicting avalanche release, the stated goal of measuring AE in snow. The authors need to provide a much more convincing case for why five different stability classes are required, and what these mean in terms of failure prediction. The

most simple case would be to have two instability classes (stable and unstable) and evaluate a simple predictive system based on crossing a β threshold value, similar to the simple forecasting model suggested by van Herwijnen et al. (2016) (and references therein). Once the β value crosses the defined threshold value, an alarm is raised for a defined duration. If an avalanche releases during this alarm, the avalanche was correctly predicted. Such a simple alarm system with two variables (β threshold value and alarm duration) could then easily be evaluated in terms of predictive performance e.g. Probability of detection and False alarm rate for a given threshold value and duration) to obtain the best suited alarm system. Of course, a more complex model could also be envisioned, provided the different instability states are clearly defined.

As suggested by the reviewer, the probable states of a snowpack built over an avalanche starting zone are defined under a separate subsection 2.5.2, as a part of the 'Methods' section. A proper justification is given why two states of snowpack are not sufficient to define the avalanche formation and release processes completely, and further the need to resolve the intermediate transitional regime of the unstable state. In the case of glacial instability evolution, Faillettaz et al. (2015) also have shown that a glacier leading towards catastrophic failure (ice avalanche) may pass through three different phases, i.e. initial stable phase, intermediate transitional phase and the regime of catastrophic failure. The avalanche formation and release processes are complex and cannot be completely defined in terms of just two the states (variables) of a snowpack, and there may be possibility of several intermediate (transitional) states of the instability evolution prior to any catastrophic failure of snowpack. In our understanding, the intermediate (transitional) regime can be subdivided into three different sub-states with respect to a critical level of instability, i.e. below the critical (sub-critical), at the critical, and above the critical (super critical) levels of instabilities; thus, making five different states including the initial stable and the last catastrophic state. Since, the transition zone is very useful in context of the avalanche formation and release where the instability changes may occur in a gradual or in a quasi-static manner; therefore, it may offer a sufficient reaction time before the release of an avalanche and could be quite helpful for avalanche prediction, well in advance.

During the very first phase of the instability evolution, a subcritical state may arise due to generation of the micro-cracks (micro-voids) under the influence of stress field. These micro-voids can be distributed randomly within the bulk, and without any specific orientation or demarcation. But during the critical instability phase, formation of the domains (clusters) of the micro-cracks start evolving within the bulk and their subsequent coalescences result into the emergence of the cracks of perceptible sizes due to more stress concentrations over such domains. Subsequently, the growth of a crack may continue until a critical crack size is achieved. The instability state corresponding to the critical size of the crack is assigned here as the super critical state of the instability. Once the critical crack size is achieved or a supercritical state has triggered, the snowpack may jump into the regime of rupture or catastrophic failure. The catastrophic failure state is highly sensitive to the internal or external stresses and may suddenly switchover to a state of the dynamic instability (chaos), in a non-linear fashion, which is rather difficult to predict using the conventional methods. The magnitude of AE activity depends on the

amount and size of the failure event which can be represented in terms of the instability index, $\beta_{in}$, derived from the AE activity; therefore, it could be a robust indicator for defining the instability states of a snowpack evolving at a particular instance. Each state can be attributed by some thresholds values of $\beta_{in}$, and may further stratify the states through a quantum jump in $\beta_{in}$-values corresponding to any transition (abrupt change) across two different states.

Through seismic monitoring and analyses of avalanche release, Herwijnen et al. (2016) have proposed simple avalanche forecasting model which is based on avalanche activity function and two tunable input parameters (alarm duration and avalanche activity threshold). An alert signal is generated following a threshold crossing corresponding to an unstable state. However, in present study, three such alert (flag) threshold signals are generated at different times corresponding to critical ($t_C$), supercritical ($t_{SC}$), and catastrophic failure ($t_{CF}$) regime. An uncertainty, in the prediction of avalanche release is primarily because of prevalence of two competitive processes such as failure and sintering in snow (Schweitzer et al., 2003; Podolskiy et al., 2014), occurring spontaneously. Therefore, an increasing trends among the flag levels after crossing the predefined threshold levels ($\beta_C$, $\beta_{SC}$, $\beta_{CF}$) would indicate the ongoing progressive damage process (initiation, nucleation, extension and arrest of the cracks) within the snowpack, in a better way with sufficient reaction time to predict the avalanche release.

Overall, the manuscript should contain more quantitative analysis of the data, beyond the rather qualitative description currently given. For instance, the authors suggest that an increase in AE count and signal strength were observed prior to avalanche release. While broadly speaking this is the case, when looking more closely at the figures it becomes clear that prior to the release of the avalanche, both the count and signal strength decreased drastically. This point is not addressed in the manuscript. Perhaps it is related to inaccuracies in the release time of the avalanche. Nevertheless, this should be discussed more thoroughly in the paper. Also, the authors did not analyse signal properties with distance to the avalanche, although this could easily be done with the array of sensors they installed.

In present case study of the avalanche formation of release event, the prominent AE parameter detected during the study period (16 Feb 2016, 11:07 am to 21 Feb 2016, 07:43 pm) are shown in Figs. 7(a, b, c, d) of revised manuscript, and earlier, in Figs. 5(a, b, c, d) of the unrevised paper. These are the original (raw) information detected in the real-time from avalanche start zone and replayed later. These figures clearly indicate that the avalanche release time, and also the times corresponding to the sharp decrease in AE counts as well as in AE signal strengths are same at 11:49 am (20 February 2016). However, after window-wise processing of the raw AE data, these timings are shifted slightly depending upon the window scale (interval), as mid of each window is shown along the time-axis for all running windows. The AE counts and signal strengths are the most prominent AE parameters which indicate distinctly (through sharp and quantum jump) the onset of critical instability and the release of avalanche. In this context, the manuscript is now been further improved with more quantified information and the issues are addressed wherever a

deviation is seen with respect to the estimated AE parameter and the observed information about avalanche release.

Further, to understand how avalanching process has progressed with respect to different sensor locations on slope, three different AE sensors (S1, S5 and S8) mounted on different AE arrestors (Ar1, Ar3, Ar4) are selected and the results so obtained are discussed within the manuscript. These sensors were selected because they are rather covering almost the entire extent of the slope (from bottom to top, and left to right). This point is further improved and addressed in the revised manuscript.

Different sensor and arrestor types were used. For me, this complicates the interpretation of the results, as it is not clear to me if the observed differences between the sensors are caused by differences in instrumentation, distance to the avalanche, or related to snow instability? These issues should be addressed more thoroughly in the paper, preferably in a quantified manner. While it is clear to me that this is perhaps difficult to do, at the very least the authors should address this point in the discussion.

We agree to reviewer that using different arrestors types would indeed complicate the interpretation. The purpose and objectives behind using two different arrestor types (cylindrical and parabolic, Table 2) were to recognize their best AE performances, also due to the reason that the present AE investigation of avalanche slope was conducted first time, and the information based on shape, capture area, etc. was unknown earlier. Interestingly, after this study, two crucial and unknown facts are learned, i.e. about the capture area of the AE arrestor and the hollow space in the interior of the arrestors. Larger the capture area of an arrestor intercepting the AE wave, more the probability of detection of AE. Secondly, snow itself is an absorbing (attenuating) medium for an AE wave, thus, by keeping one surface of arrestor free from snow (without any contact to snow), by keeping interior space hollow, the AE oscillations (plate-wave oscillation) can sustain longer within the arrestor medium. This is crucial information to design an AE arrestor. In present study, two types of the arrestors were used to detect the AE, the cylindrical arrestors (Ar1, Ar2, Ar4, Ar5) and parabolic AE arrestors (Ar3, Ar6), where none of the sensors mounted on Ar6 (S11, S12) and other sensors S3, S6 and S9 were out of order during the study period. The responses of both the sensors (S4, S5) mounted on the parabolic arrestor (Ar3) was observed relatively poorer as compared to other sensors. It is possibly because of the smaller capture area of the parabolic arrestor (Ar3 and also because of the both surfaces of Ar3 being in the contact of snow (Fig. 2b). This point is further addressed in the discussion.

The overall structure of the paper is not ideal. There is no clear and succinct description of the site (e.g. Figure 3 should also include a map showing the location of the site), instrumentation and the methods. For instance, the authors mention that the AE sensors were deployed on an avalanche slope. However, it is not specified when and how this was done, and at what depth in the snow cover the sensors were placed. Furthermore, results and discussion are continuously mixed, and the text is often repetitive. The authors should restructure the manuscript and

improve the readability. Furthermore, there are many grammatical errors throughout the paper (for instance the first two sentences of the abstract), and I would encourage the authors to seek the help of a native English speaker to go over the manuscript before resubmitting it.

As suggested by the reviewer, the study site is shown through its location on map (now Fig. 1a) along with its geographical coordinates (latitude, longitude and elevation). The structure of the manuscript has now been improved significantly. An accessible avalanche slope was selected near Patsio (in Great Himalayan region of India) with a viable AE instrumentation in order to have a continuous watch on the slope. The details about the instrumentation, along with their deployment on avalanche start zone, are addressed clearly in different sub-sections of the method section. In present setup, the arrestors (cylindrical and parabolic) were fixed in the bottom layer of the snowpack (during the onset of the winter season, when snow height was about 0.3 m) and were not anchored to the ground. The purpose behind such mounting was basically to get the arrestors submerged well in the snow medium, without any ground anchorage. The top of the arrestor was covered by a lid and bottom was open to ground side. The AE sensors were fixed to the arrestor, once it had settled well in snow while snow height was about 30 cm; half of the arrestors were buried in snow and half in air. The AE sensors were coupled to the arrestor surface using specialized couplant (high vacuum silicon grease). Further, a housing arrangement was provided to safe guard the sensor and also preventing any kinds of the stray movements relative to the arrestor surface which may otherwise generate undesired signal (noise). The AE sensors were connected to a broadband multichannel AE system via high grade (lossless) coaxial cables of length 180m. After installation of the AE arrestor, sensor, system, the AE data acquisition was started after 28 January 2016 onwards and continued for the entire winter season. The pertaining section has been improved further in regard to above information.

Throughout the paper, there are numerous instances of inaccurate wording, poorly defined terminology and many unfounded statements. For instance, the authors write that changes in $\beta$ correlated to progressive failure / damage (e.g. lines 437-438; lines 484-485; lines 486-487). However, failure or damage was not measured, and it is thus not possible to correlate both. In this context, it is also not clear how the stability threshold values in Figure 9 were determined. The authors should provide a description of the method/criteria used for this.

The manuscript has been further revised, also in regard to the language and scientific components, terminology, grammar, etc. The repetitions found, at many places, within the manuscript are removed, and inaccurate wordings are corrected, accordingly.

Here, authors have attempted to link the abruptly increasing AE activity to the progressive damage process that might have occurred within the weak layer of the snowpack below a cohesive slab that ultimately led to the imminent failure of the snowpack (observed in the form of avalanche release). In the authors understanding, the correlation between the increased AE activities and the progressive damage process of the snowpack (weak layer) has been established through the reported studies and the measured facts. In this study, two

measurable facts are noted, an abrupt jump in the AE activity (shown by all the sensor deployed at different locations on avalanche starting zone) followed by a linearly increasing trend of AE leading towards a maximum value and dropped sharply thereafter. Secondly, the measured time of avalanche release. But interestingly, the time of AE maxima and the actual time of avalanche release are matching together. Further, from past studies of numerous researchers (Schweizer et al., 2003; McClung, 2011; Reiweger et al., 2015; Reuter et al., 2015; Schweizer et al., 2016; Gaume et al., 2017; Capelli, 2018), these facts are established now as main contributory factors behind an avalanche release, the presence of a weak layer below a cohesive slab, and a progressive damage process prior to the imminent failure of a slab. If avalanche release has been confirmed in the present case, then it looks logical to say that it has certainly passed through the course of progressive damages before the catastrophic failure of the slab. A link between the abrupt and linearly increasing AE activity and the progressive damage process within the snowpack through the course of instability evolution can thus be helpful to determine the instability states where quantum jumps in the $\beta_{in}$ - values can be seen for any transition across to different states. However, in any particular state the $\beta_{in}$ – plot may vary or fluctuate in some small range of $\beta_{in}$. In regard to above explanation/justification, the manuscript has been improved further, wherever is discussed.

The presentation of the data and the results is not very convincing. For instance, the authors should show a figure with the relevant meteorological data (snow height, air temperature and wind speed) of the study period. The authors should also show the snow profile graphically (not a table), show stability test results and the location of the failure layer, and indicate the location of the avalanche failure layer. Figure 4 should show the avalanche that released on the study site. However, I do not see any avalanche debris confirming the release of the avalanche, and the quality of the image is rather poor. Finally, Figure 7 and 8 could be combined. Indeed, for each sensor, the large window average shown in Figure 7 could be plotted on top of the shorter window values shown in Figure 8.

As suggested by the reviewer, the relevant meteorological parameters, collected during the study period, are now presented in the graphical form, shown in Figs. 5 (a, b). Also, the snow profile is drawn in the graphical format (Fig. 6). The location of the failure layers along with the stability test results are marked over this figure, as suggested by reviewer. We do agree that the image quality of avalanche was poor now this image is replaced by an image of better quality (Fig. 4), showing the avalanche debris and other features. The Figs. 7 and 8 are clubbed together (now shown in Fig. 9) as suggested by reviewer.

Finally, the authors show data from one snowfall and the subsequent avalanche. It would be very interesting to show data from an additional snowfall without an avalanche, if these data exist. This would provide more insight into the robustness of the observations. Are the results presented in the paper associated with avalanche release, or are the same characteristics also

observed for other snowfall events without an avalanche? At the very least, the authors should address this point in the discussion, and more generally the limitations of their results.

Authors agree to the reviewer that the robustness of the information would be better conveyed by showing the effect of snowfall on AE activity particularly during the non-avalanche period. The effect of snowfall on AE from an unstable snowpack (with avalanche) is already provided under the Figs. 7(a, b, c, d). However, to know the effect of snowfall on a stable snowpack (without avalanche), we are presenting the AE data for a period of over 12 hours detected during the period 8:44 am to 9.18 pm of 17 March 2016 from the same slope with same set of the instrumentation. In this region, the snowfall has started around 11:30 am of 17 March 2016, and it continued until 1:30 pm of 19 March 2016. However, between 11:30 am to 9:18 pm of 17 March 2016, the snow fall rate varied from 4 mm/hour to 20 mm/hour with total amount of snowfall was received was 105 mm. Also, the wind speed was observed varying from 2.2 to 6.5 m/s. Further, the air temperature during this period was varied from -2.0 to $-5.0^0$C, and the snow surface temperature from $- 7.5^0$C to $-9.5^0$C. A one-to-one correlation between the snowfall rate and the AE activity (in terms of AE counts, amplitude, signal strength and AE energy are presented under Appendix-A through Figs. A1(a, b, c, d). Since the size of the paper seems to be considerably large, therefore, authors feel to keep this section under the Appendix-A. The above observations has clearly indicated that the effect of snowfall is least on the AE activity detected from a stable snowpack or an unstable snowpack, provided the arrestors surfaces are not directly hit by the snow particles. Above information is also addressed within the revised manuscript.

**4.5 Prediction of Avalanche release from temporal evolution of $\beta_{in}$**

As discussed in previous sections, the sharp increase in the AE characteristics, and also in the $\beta_{in}$ – values (observed nearly 13 hours before the avalanche release), following almost a linearly increasing trend thereafter, and approaching towards the imminent failure of the snowpack, could be useful information for development of an avalanche prediction model. Further, after the onset of increased (quantum jump) AE activities could possibly be because of the triggering of the critical instability within the snowpack. The progressive growth pattern of $\beta_{in}$ after onset of the critical instability can be related to the ongoing damage or crack extension processes occurring within the weak layer of a snowpack. At a particular instant ($t_i$), a snowpack may possess a particular state (stable or unstable) and can be attributed by some value of $\beta_{in}$ ($t_i$). Based upon our observations (Figs. 11) on progressive linear increase for $\beta_{in}$, registered from the avalanche slope, the temporal evolution of $\beta_{in}$ can be represented by, $\beta_{in}(t_i) = \lambda t_i + \zeta$, where $\lambda$ and $\zeta$ are the constants for instability evolution; the $\lambda$ represents how fast the instability is changing, and the $\zeta$ is an offset level prior to the onset of critical instability. Since, the linearity in $\beta_{in}$ ($t_i$) is only between the to onset times for critical ($t_C$) and super critical ($t_{SC}$) states of snowpack instability, therefore, the relation $\beta_{in}(t_i) = \lambda t_i + \zeta$ is valid only for the region, $t_C < t_i < t_{SC}$. If $\beta_C$ and $\beta_{SC}$ are the instability indices corresponding to $t_C$ and $t_{SC}$, respectively, then $\lambda$ and $\zeta$ can be determined, $\lambda = \left( \frac{\beta_{SC} - \beta_C}{t_{SC} - t_C} \right)$ and $\zeta = \left( \frac{\beta_{SC} \, t_C - \beta_C \, t_{SC}}{t_{SC} - t_C} \right)$.

In general, the solution of the state function $\phi$ ($t_i$, $\beta_{in}(t_i)$) beyond the supercritical state of the snowpack and in the catastrophic regime could be used to predict the imminent failure of the snowpack (avalanche release). For, region $t_i > t_{SC}$, the $\beta_{in}(t_i)$ may not strictly follow the linearity because of jumping to a dynamic (disordered) state of instability from quasi-static or static states. However, in a simplest form, the state function $\phi$ ($t_i$, $\beta_{in}(t_i)$) could be extrapolated in the failure regime (beyond $t_{SC}$) to predict the imminent failure. The catastrophic failure ($t_{CF}$) of the snowpack can thus be determined from the threshold values of $\beta_{in}$ for critical ($\beta_C$), super critical ($\beta_{SC}$) and catastrophic failure ($\beta_{CF}$) states. From above information, the onset of a super critical state can therefore be predicted by with respect to the onset of critical state by,

$$t_{SC} = \left( \frac{(\beta_{SC} - \beta_c) t_i - ((\beta_{SC} - \beta(t_i)) t_C}{\beta(t_i) - \beta_C} \right).$$

The catastrophic failure time ($t_{CF}$), with respect to the supercritical state ($t_{SC}$), can be approximated by, $\quad t_{CF} = t_{SC} + \left( \frac{\beta_{CF} - \beta_{SC}}{\lambda_f} \right)$ \hfill (3)

Here, $\lambda_f$ is a factor deciding the behaviour of the catastrophic (dynamic) regime. This model can be used to predict the failure of a snowpack or avalanche release and the input parameters to this model requires the information on present instability ($t_i$, $\beta_{in}$ ($t_i$)) condition, the critical state parameters ($t_C$, $\beta_C$), and the threshold values for $\beta_{SC}$ and $\beta_{CF}$. In this case, the triggering time corresponding to super critical state ($t_{SC}$) is not necessarily required to predict the failure, therefore, it may be helpful particularly in those cases where there is not sufficient reaction time between the super critical transition and the catastrophic failure of the snowpack. In present study, the threshold values for $\beta_C$, $\beta_{SC}$ and $\beta_{CF}$ are estimated based on one successful avalanche release event (Table 5, now Table 3). From our database, the estimated values of $\beta_C$ has a range of 0.65 to 0.73 through small to large window scales. Similarly, the $\beta_{SC}$ and $\beta_{CF}$ were estimated in the range of 1.04 to 1.28, and 1.14 to 1.36, respectively. The values of $\beta_C$ may further improved through observations of other cases of avalanche occurrences and corresponding to any unknown scatter of $\beta_{in}$ ($t_i$), where a sudden jump in $\beta_{in}$ – values followed a linearly increasing behaviour is observed. However, above values $\beta_{SC}$ and $\beta_{CF}$ could be considered as reference levels for any unknown case. For implementation of the prediction model, it is also essential to have the understanding about the parameter, $\lambda_f$, as used in Equation (3). The $\lambda_f$ decides how fast a supercritical state switches over to a dynamic state of the failure, where non-linearity would be experienced for $\beta_{in}$. The value of $\lambda_f$ may vary between cases to case. For computational purposes, the values of $\lambda_f$ are approximated from 1.37 x 10 s$^{-1}$ to 1.56 x10$^{-5}$ s$^{-1}$, through small to large window analyses.

Using above parametric information, the model (Equation 3) was tested to predict failure times ($t_{CF}$) of snowpack for $t_C < t_i < t_{SC}$. The results are presented in Figs. 11 (a, b) for large as well as small window analyses where AE data corresponding to four different and best responding AE sensors ($S_1$, $S_2$, $S_7$ and $S_8$) are selected for prediction of the snowpack failure. In present case of avalanche release event, the actual failure time of the snowpack was registered to be 3,48,283 s, therefore, the performance of the model would be better if the predicted failure time is close or overlapping to the observed failure time. In this study, the running time scale is considered in seconds for sake of easier computations in linearly increasing manner. In Fig. 11(a, b), $t_{CF}$ (predicted) values are plotted along the y-axis which are generated from $\beta_{in}(t_i)$ data set corresponding to the present (running) time, $t_i$, and are randomly selected from large scatter of $\beta_{in}$ ($t_i$) database for $t_C < t_i < t_{SC}$. Further, the deviations (%) of $t_{CF}$ (predicted) with respect to the actual release time of the avalanche, i.e. [$t_{CF}$ (measured) - $t_{CF}$ (predicted)] /$t_{CF}$ (measured), are also plotted corresponding to different sensors and window scales, and results are shown in Figs. 11(c, d). Through a least square fit of the data, the correlation coefficient ($R^2$) is also estimated for different sensors and their values are shown over the figure itself. From the large window analysis that the sensor $S_2$ and $S_8$ have predicted well (> 90% value closer to the observed failure time), i. e. $R^2$ ($S_2$) = 0.90, $R^2$ ($S_8$) = 0.919; however, sensor $S_1$ has not predicted as expected, i.e. $R^2$ ($S_1$) = 0.331. But in the case of $S_7$, a large scatter was observed in the predicted values. But for small scale window analysis, the sensor $S_1$ has predicted better and closer to the expected value of $t_{CF}$, i.e. $R^2$ ($S_1$) = 0.875; however, the prediction of $S_2$ and $S_7$ are little poor, i. e. $R^2$ ($S_2$)

= 0.517, $R^2$ ($S_7$) = 0.544. For small window analysis, the prediction of $S_8$ is found poorest among all. At some instances, over predicted values are also generated by these sensors. From this analysis, it is found that the deviation in prediction is large in the beginning but decreasing gradually as moving closer towards the observed failure time. These results also indicate that as the present time, $t_i$, (where forecast is generated) is approaching towards the observed failure time; the accuracy in the prediction is increasing significantly irrespective of the sensor.

**5.5 Prediction of Avalanche release from temporal evolution of $\beta_{in}$**

The prediction of avalanche release, directly in terms of the AE parameters, has been a longstanding research area and to the best of our knowledge, hardly any such report has been published in the recent past. However, in cases of several materials and structures, efforts have been made to develop the failure prediction model. Voight (1989) developed a simple model for rate-dependent failure of a material, directly in terms of a measurable quantity such as strain ($\varepsilon$). The failure time ($t_f$) could be approximated by $t_f - t_i = \frac{1}{A\dot{\varepsilon}_i}$ where $t_i$ is arbitrary time, A is constant corresponding to terminal stage of the failure, and $\varepsilon_i$ is the arbitrary strain rate. This relation is quite useful in the sense that the failure of the materials can be predicted in terms of present (arbitrary) state variables. For prediction of the catastrophic failures of the glaciers, Faillettaz et al. (2015) have presented a power law accelerations for surface displacements through a model for temporal evolution of the surface displacements, i.e. d(t) = A$_0$ + d$_s$t − B(t$_c$ − t)$^m$ f′(t)) where d$_s$ is continuous displacement, t$_c$ is the critical time, f′(t)) is function of periodic oscillations, and m < 1 is an exponent of power law; here, A$_0$ and B are some arbitrary constants. This model has been used to predict of the glacial break off. Further, for prediction of snow failure, a linear regression relation ($b = \alpha(t_f - t_i)/t_f$, where $\alpha$ is a constant and t$_f$ is failure time) is tested by Capelli et al. (2018) through evolution of b-value (exponent of power law) derived from AE. They have shown an abrupt decrease in the b-values corresponding to the failure of snow samples passed under different loading conditions. In addition, van Herwijnen et al. (2016) have analysed the seismic monitoring of the avalanche release and proposed a simple avalanche forecast model using a function termed as avalanche activity function ($A(t, d) = A(t, d)$ +1 for t$_i$ <$t$ < t$_i$ +d.). This model utilizes just two tunable parameters, i.e. alarm duration (d) and avalanche activity threshold, and thus an alarm can be generated once avalanche activity function exceeds a particular threshold level.

      In present work, a different approach is applied for prediction of snowpack failure on a slope (avalanche release) through temporal evolution of the instability index, $\beta_{in}$, derived from AE counts and amplitude. The observed abrupt increase in AE activity, nearly 13 hours prior to the avalanche release, and thereafter a linearly increasing trend of $\beta_{in}$ leading towards the imminent failure of the snowpack is applied to predict the avalanche release. A model for failure prediction of the snowpack is proposed through a state function for arbitrary value of $\beta_{in}$ (t$_i$) measured at a time, t$_i$. From observed data, the onset of a supercritical state (t$_{SC}$) is predictable through a linear regression relation for $\beta_{in}$ for interval t$_C$ < t$_i$ < t$_{SC}$; however, the failure time could be predicted by extrapolating it in the failure regime which can be characterized by the parameter, $\lambda_f$. In proposed prediction model, the input parameters are primarily the threshold values of $\beta_{in}$ pertaining to the critical, super critical and the failure state of the snowpack. If present instability status ($\beta_{in}$(t$_i$)) of the snowpack is known (t$_C$ < t$_i$< t$_{SC}$) after triggering of critical instability, the failure could be predicted with respect to the onset of critical level (t$_c$). The model

performance is tested (Fig. 11) with respect to different sensors (installed at different locations on avalanche slope) and for two window scales. Since, this is a case study based on continuous AE monitoring of avalanche slope and being reported for first time through registration of one successful avalanche release event. The AE activity recorded by four different sensors was used to test the model following the onset of critical instability but different accuracies in the failure prediction are found for different sensors. The best prediction accuracy was achieved over 90% but it is not consistent to the window scales. For a snowpack approaching towards a catastrophic failure, all predicted values generated at any instance should always converge to the actual failure time ($t_{CF}$). The success or performance of a prediction model would more depend upon the measure of convergence about $t_{CF}$. But in our analysis, the convergence in the prediction is certainly there but it is better as $t_i$ is closer to actual failure time ($t_{CF}$). The possibility of errors in the poor prediction by some of the sensor could be due to random selection of $t_i$, from a large AE data volume (recorded in the scales of micro-seconds) and thus having a lot of fluctuations in the instantaneous values of $\beta_{in}$ ($t_i$). The approximation in $\lambda_f$ might also restrict the prediction accuracy. These predictions are preliminary results in the direction of model development using $\beta_{in}$ - scatter but an essential step towards development of AE based warning for avalanche release. The prediction accuracy would indeed enhance through multiple level model testing and fine tuning of output by large sets of AE database collected for avalanche release.

**Appendix A: AE data set recorded during snowfall with no avalanche**

In order to justify the abnormal AE activity (shown in fig. 7) prior to avalanche release, which was independent of major effect of snowfall and wind activity, a different AE data set is presented in this section. This data set was obtained during snowfall but no avalanching was recorded afterwards. The AE data recorded from 08:44 hrs of 17[th] March 2016 to 00:00 hrs of 18[th] March 2016 is presented in terms of AE counts, amplitude, signal strength and frequency centroid in Fig. A1(a, b, c, d) respectively. The shaded region indicates the snowfall rate in mm/hour. The rate of snowfall varied from 4 – 20 mm/hour and the total amount of snow received during this period is 105 mm. Also, the wind speed was measured in the range of 2.2 to 6.5 m/s. From fig. A1, it can be inferred that no specific increase is observed in AE activity during snowfall. In fact, a random scatter in terms of AE counts and amplitude is observed, where the maximum values reported here are 2207 in Fig. A1(a) and 87 dB in Fig. A1(b) respectively. Similarly, signal strength and frequency centroid depict a low level AE activity. On contrary, the maximum values of AE parameters recorded during the avalanche formation period are quite high and thus indicate the process of instability development prior to avalanche release.

[Figure]

**Figure A1:** AE characteristics registered during 08:44 hrs of 17[th] March 2016 to 00:00 hrs of 18[th] March 2016 for sensor S1, (a) AE-Counts, (b) AE-Amplitude, (c) Signal Strength, (d) Frequency Centroid. In all the plots, the shaded region represents the rate of snowfall (mm/hour).

**Revised Figures with captions**

**Figure 1:** Location of the Study Site (Avalanche Starting Zone), near Patsio in Great Himalayan range of India (latitude: 32°45′11.16″N, longitude: 77° 15′ 37.88″E, elevation 3794m)

[Figure]

**Figure 2:** (a) Top side view of the cylindrical AE Arrestors and the AE sensor Housing, (b) Parabolic AE Arrestor (with its sensor housing) partially sub-merged into snow

[Figure]

[Figure]

**Figure 3:** A typical waveform representing the extents of AE counts and amplitude, derived from a voltage signal of an AE-hit. In this figure the black dotted curves (fluctuations) are representing the AE counts (the numbers of ripples above the threshold level)

[Figure]

**Figure 4:** (a) Front view of the avalanche slope showing locations of different AE arrestors (Ar1 to Ar6) relative to the respective pole (P1 to P7) positions on slope (the poles of height 3.0m are erected on slope for locating the arrestors during winter after they get fully buried in snow, (b) Front view of the avalanche slope after release of the avalanche reported at 11:49 am of 20 Feb 2016. The red dotted line is showing the fracture line and the P1 to P7 are poles used to locate the arrestors after they get fully buried in snow.

[Figure]

**Figure 5:** (a) Meteorological parameters collected from the study site for period 15 Feb 2016, (8:30 am) to 22 Feb 2016 (5:30 PM), in terms of wind speed, air temperature and snow surface temperature with respect to the snow fall rate shown by the shaded region. (b) Meteorological parameters collected from the study site for period 15 Feb 2016 (8:30 am) to 22 Feb 2016 (5:30 PM), in terms of standing snow, relative humidity and atmospheric pressure with respect to the snow fall rate shown by shaded region.

[Figure]

[Figure]

**Figure 6:** Snowpack profile collected post avalanche release through after the pit-analysis. Different snow layers including the weak layer relative to the failed weak layer positions along with their compositions and that of the weak layer (DH crystal) are depicted on it.

[Figure]

**Figure 7:** Prominent AE characteristics registered during avalanche formation and release processes for sensor S1, (a) AE-Counts, (b) AE-Amplitude, (c) Signal Strength, (d) Frequency Centroid (the start AE acquisition time is 16 Feb 2016 at 11:07 am and the end of AE acquisition time is 21 February 2016 at 07:43 pm). The yellow line in respective figure represents the mean values of the recorded AE parameters estimated over running window of interval 600 s. The two different scales for time (along x-axis) are shown for experimental time in terms of running seconds and also with respect to the local time in IST (Indian Standard Time) and both the scales are synchronized. In all the plots, the shaded region represents the snowfall period (start of snowfall, 19 Feb 2016 at 07:10 am and end of snowfall, 20 Feb 2016 at 10:20 am) and dotted line indicates the time of avalanche occurrence (20 February 2016 at 11:49 am).

[Figure]

**Figure 8.**Window-wise (w: 600 s) variation in some prominent AE characteristics captured by AE-Sensors S1, S5 and S8 (placed in slope at different locations) during avalanche formation process, (a) Max-value of Hit Duration, (b) Sum-value of AE-Energy, (c) Max value of ASL and (d) Max value of RMS. The dotted line indicates avalanche occurrence time (20 February 2016 at 11:49 am). The saturation observed in the case of AE hit duration (Fig. 8(a)) is due to the limitation of AE acquisition setup after attaining the maximum level of hit duration (1 s).

[Figure]

[Figure]

[Figure]

**Figure 9**. Variation in instability index corresponding to different AE Sensors placed on slope, (a) Sensor S1, (b) Sensor S2, (c) Sensor S4 , (d) Sensor S5, (e) Sensor S7, (f) Sensor S8. Here, the variation in instability index $\beta_{in}$(lw) corresponding to the larger window of variable size (w: 26 minutes to 96 minutes) is shown by thick black line with red dots and $\beta_{in}$(sw) corresponding to the smaller window size (w: 60s) is shown by dots. The shaded region indicates the snowfall rate (mm/hour) and the dotted line indicates the time of avalanche release.

[Figure]

**Figure 10.** Identification of critical transition states for snowpack instability during avalanche formation by monitoring the temporal variation of $\beta_{in}$ corresponding to (a) large (variable) window sizes (w: 26 minutes to 96 minutes), (b) smaller window of fixed interval (w: 60 s) for critical period of avalanche formation. The shaded region indicates the snowfall period and the dotted line the time of avalanche release.

[Figure]

[Figure]

**Figure 11.** Prediction of Snowpack failure model to generate the predicted failure times ($t_{CF}$) w.r.t. $t_i$, corresponding to four different AE sensors (S1, S2, S7 and S8) is presented (a) for large window size (w: 26 minutes to 96 minutes), (b) for small window size (w: 60s) and the prediction deviations (%) of predicted failure times ($t_{CF}$) with respect to the actual release time of the avalanche are shown (c) for large window size (w: 26 minutes to 96 minutes), (d) for small window size (w: 60s)

[Figure]

[Figure]

[Figure]

[Figure]

**Additional References:**

1. Capelli, A., Reiweger, I., Schweizer, J.: Acoustic emission signatures prior to snow failure, Journal of Glaciology, May (2018), doi: 10.1017/jog.2018.43, 2018.\\

2. Faillettaz, J., Funk, M. and Vincent, C.: Avalanching glacier instabilities: Review on processes and early warning perspectives. Rev. Geophy., 53(2): 203-224, doi:10.1002/2014RG000466, 2015.

3. Flotron, A.: Movement studies on hanging glaciers in relation with an ice avalanche, *J. Glaciol.*, *19*(81), 671–672,1977.

4. McClung, D. M.: Analysis of critical length measurements for dry snow slab weak-layer shear fracture J. Glaciol., 57(203), 557-566, 2011.

5. Pralong, A. and Funk, M.: On the instability of avalanching glaciers, *J. Glaciol.*, *52*(176), 31–48, doi:10.3189/172756506781828980, 2006.

---

## Author Comment (AC2) · 9 Jun 2020

**Reply to the comments from Referee 2**

**Acoustic Emission investigation for avalanche formation and release: A case study of dry-slab avalanche event in Great Himalaya by Kapil et al. (tc-2020-38)**

Below are the replies (in 'blue' text) from authors against the comments made by the reviewer (in 'black' texts) on above manuscript.

**Comment (Ref 2):**

I found the paper to be too long for the subject matter. I appreciate the efforts of using AE technique towards avalanche mitigation processes. It's a good tool to have; no doubt. The authors, however, should have refrained from making comments with buzzwords, again and again, on micro mechanisms because they seem to be not aware of the "fundamental fact" that snow is an extremely "High-temperature material" undergoing stress-temperature-time induced morphological changes. The micro- and macro failure mechanisms and their kinetics are very complicated. Nonetheless, significant progresses has been made in the past on physics of failure in polycrystalline solids (not necessarily porous media) at high-homologous temperatures, close to 0.99 Tm, but snow is porous and the authors do not seem to be aware of those developments. For example, there were AE studies in pure ice for examining the microstructure property relations - way back in the early 1980's. Specifically, they looked at polycrystalline ice as a "high-temperature" material (existing in our cryosphere at extremely high homologous temperatures, higher than about 0.9 Tm, where Tm is the melting point in Kelvin). Old ideas on AE, based on micro-mechanisms in other engineering or geological materials at low homologous temperatures should be discarded. However, the use of AE technology can still be used as a tool - such as monitoring the snow states. The authors tried to use AE for real practical application - and should have stayed in that arena - instead of going beyond the realm of the data and speculating about dislocations, etc. They seem to impress the audience with mechanisms that they actually do not fully understand. The manuscript, in this regard, should be modified significantly to bring out the real strength of the work.

**Reply:**

Authors sincerely thank to reviewer for his insightful comments on our manuscript submitted to 'The Cryosphere'. We do agree that the manuscript is lengthy; it is because authors wanted to report most of the relevant information collected during a natural avalanche release event, registered successfully for the first time, through acoustic emission (AE) monitoring of the avalanche starting zone. Authors agree to reviewer that certain aspects like the failure of polycrystalline ice at higher homologous temperature were not covered and such information would indeed improve the quality of the work done. Authors have cited the published literature in contexts of recent understanding deformation and failure behaviour of polycrystalline ice particularly at higher homologous temperatures, dislocation dynamics relative to migration of ice grain boundaries, the mechanism of slab failure, and are discussed within the revised manuscript. In response to reviewer's comments the manuscript has been modified and improved significantly. The buzz-words are removed and corrected.

The deformation characteristics of the snow grains define the mechanical behaviour of a snowpack which is an aggregate of polycrystalline ice grains (Sommerfeld, 1970). The changes in snow morphology and structure are quite sensitive to temperature and load (McClung, 1996); therefore, the deformation and failure behaviour of polycrystalline ice can provide some key insights about the failure mechanism of a snowpack as polycrystalline ice is a constituent of snow itself. In present study, the snow (surface) temperature was observed varying from $-2.1^0$C to $-2.5^0$C ($0.92 T_m$ to $0.99 T_m$) during the study period; therefore, the deformation behaviour of the snowpack would be affected by prevailing temperature conditions, and correspondingly to the released AE. In fact, the process of AE generation and its propagation is a complex process within a snowpack; however, it is essential to correlate the released AE to the deformation and failure processes occurring from microscopic to macroscopic scales during formation and release of an avalanche. Sinha (1978) related the deformation behaviour of polycrystalline ice as a function of temperature, time, and stress through phenomenological relation for creep. The crystallographic structure, direction of application of load and strain history of ice affects the deformation behaviour of polycrystalline ice. At high homologous temperature ($>0.9 T_m$), Sinha (1984) has shown that the grain boundary sliding and delayed elasticity can initiate the micro-cracking of polycrystalline ice and once a critical state stress is achieved, the cracking in ice may occur depending upon temperature but independent of the grain size. The dislocation dynamics of deforming snow represents the strain bursts in which the grain boundary deformation mediates the dislocation stress fields to nearby grains. The deformation and failure of a snowpack resulted from the dislocation could be manifested by stress waves (AE) propagating within the weak layer.

The grain boundary (GB) interaction during creep may act as a source of lattice dislocations and an obstacle to the dislocation movement whereas the dislocations can be generated both from free surface GB intersections and from the interiors of GBs (Liu et al., 1995). The strain in ice may tend to concentrate near the grain boundaries (GB) and sub-grain boundaries (SGB) in relation to the stress field heterogeneities, and a strong connection between nucleation and local internal stress field (Grennerat et al., 2012) is inferred from dislocation arrangements. The high temperature ($T = -5°C$, $\sigma = 0.5$ MPa) creep response of polycrystalline columnar ice is shown by Chauve et al., (2017) where strain heterogeneity may result in strain induced grain boundary migration (SIBM). The nucleation mechanism in ice occurs due to grain boundary bulging as a result of SIBM. The tilt SGBs or kink bands composed of basal edge dislocations are commonly observed in deformed poly-crystalline ice. The ductile-to-brittle transition is an effect of constant striving among crack-tip creep and crack propagation. The cohesive zone models are also applied to simulate the dynamics of fracture during crack development in ice (Gribanov et al., 2018). Journaux et al. (2019) have characterised the stress strain heterogeneities and deformation mechanisms in polycrystalline ice at high temperature. They proposed the nucleation by bulging, sub-grain boundary formation followed by grain growth to explain the evolution of crystallographic preferred orientation (CPO) during dynamic recrystallization. The AE response of creep cracking in polycrystalline ice at high homologous temperature $> 0.96 T_m$ was studied by Sinha (1996) where decelerating primary creep followed by accelerating tertiary creep effects were seen in relation to AE. The AE behaviour of creeping ice is investigated by Richeton et al. (2005) and

they have shown that plasticity in polycrystalline ice is characterized by intermittency in the dislocation avalanches but the grain boundaries may hinder the avalanche propagation.

Snow exhibits characteristic AE responses under different processes such as dislocation movements, plastic deformation, breakage of bonds and grain ruptures (St. Lawrence, 1980). The dislocation dynamics, a critical phenomenon for AE generation, is useful to understand the plastic deformation complexity of crystalline materials (Weiss et al., 2000) which was further related to AEs for visco-elastic materials (Miguel et al. 2001). For the release of a dry-snow slab avalanche, the failure of a weak layer below a cohesive slab is necessary and sufficient condition and the crack would propagate across a slope once the crack length exceeds a critical size which is almost independent of slope angle for crack propagation (Gaume et al., 2017) and the avalanche release is attributed to the formation of shear bands as an outcome of localization of micro-fractures during the deformation of snow (McClung, 1981). The pattern of AE observed during shearing is suggestive of slip surface formation in snow which acts as stress concentrators even in absence of natural imperfections (McClung, 1987). In a snowpack, the unstable conditions may prevail until a weak layer exist below a cohesive slab, and release of avalanche is associated to the initiation and propagation of the cracks within the weak layer (Schweizer et al., 2003; Gaume et al., 2018). To explain the dry slab failure at high homologous temperature ($> 0.90$ $T_m$), micromechanical models were applied by Schweizer et al. (2003), where two competing processes such as damage and sintering collectively affecting the snow deformation and slab rupture. Recently, Gaume et al. (2018) have presented the volumetric collapse of a cohesive slab leading to the localization of compacting shear bands following the anticrack propagation. A pre-existing shear stress across thin weak layer could be a significant factor deciding the fracture of a slab (Bazant et al., 2003). For release of an avalanche, the critical crack length is essential before any shear fracture occurs, McClung (2011) estimated the ratio of critical crack length and slab depth to vary from 0.1 to 2.0. For quantitative assessment of snow instability, the crack propagation propensity (Schweizer et al., 2016; Reuter and Schweizer, 2018) is considered as a measure of complex slab-weak layer interaction. The total amount of mechanical energy supplied during loading of a snowpack may result into the increase of its internal energy, fracture surface energy, dissipation (viscous and thermal) energy and kinetic energy. A new fracture surface can be created at the cost of the free surface energy which can be contributed by stored internal energy of the system, and a part of the fracture surface energy could be experienced in terms of the AE.

**Additional References:**

1.  Bazant Z. P., Zi G. and McClung D.: Size effect law and fracture mechanics of the triggering of dry snow slab avalanches, J. Geophys. Res., 108 (B2), 2119, 13-1 to 13-11. doi:10.1029/2002JB001884, 2003.
2.  Chauve T., Montagnat M., Barou F., Hidas K., Tommasi A., and Mainprice D.: Investigation of nucleation processes during dynamic recrystallization of ice using cryo-EBSD, Philos. Trans A Math Phys Eng Sci., 375(2086): 20150345. doi: 10.1098/rsta.2015.0345, 2017.

3. Gaume J., van Herwijnen A., Chambon G., Wever N., and Schweizer J. Snow fracture in relation to slab avalanche release: critical state for the onset of crack propagation, The Cryosphere, 11, 217–228, 2017, doi:10.5194/tc-11-217-2017, 2017.

4. Gaume J., Gast T., Teran J., van Herwijnen A. and Jiang C.: Dynamic anticrack propagation in snow, Nature Communications 9:3047, doi: 10.1038/s41467-018-05181-w., 1-10, 2018.

5. Grennerat F, Montagnat M, Castelnau O, Vacher P, Moulinec H, Suquet P, Duval P.: Experimental characterization of the intragranular strain field in columnar ice during transient creep. *Acta Mater.* 60, 3655–3666, doi:10.1016/j.actamat.2012.03.025, 2012.

6. Gribanov, I., Taylor,R., and Sarracino, R.: Cohesive zone micromechanical model for compressive and tensile failure of polycrystalline ice, Engineering Fracture Mechanics, 196(1), 142-156,DOI: 10.1016/j.engfracmech.2018.04.023, 2018

7. Journaux B., Chauve T., Montagnat M., Tommasi A., Barou F., Mainprice D. and Gest L. Recrystallization processes, microstructure and crystallographic preferred orientation evolution in polycrystalline ice during high-temperature simple shear, The Cryosphere, 13, 1495–1511, 2019. https://doi.org/10.5194/tc-13-1495-2019.

8. Liu F., Thayer I. B., and Dudle M.: Dislocation-grain boundary interactions in ice crystals, Phylosophical Magazine A, 71(1), 15-42, 1995.

9. McClung, D.M.: Failure characteristics of Alpine snow in slope deformation, Proc. Int. Symp. mech. Behav. of struc. Media, Ottawa, Canada 409-418, 1981.

10. McClung D.M.: Mechanics of snow slab failure from a geotechnical perspective Avalanche Formation, Movement and Effects, Proc. of Davos Symp. IAHS Publ. No. 162, 475-507, 1987.

11. McClung D. M.: Effects of temperature on fracture in dry slab avalanche release, *J. Geophys. Res.*, *101*(B10), 21907–21920, 1996.

12. McClung D.M.: Analysis of critical length measurements for dry snow slab weak-layer shear fracture, Journal of Glaciology, Vol. 57, No. 203, 557-566, 2011.

13. Reuter B., and Schweizer J.: Describing Snow Instability by Failure Initiation, Crack Propagation, and Slab Tensile Support, Geophys. Res. Letters, 45, 7019–7027. https://doi.org/ 10.1029/2018GL078069, 2018.

14. Richeton T., Weiss J., and Louchet F.: Breakdown of avalanche critical behaviour in polycrystalline plasticity, Nature Materials, 4, 465–469, 2005.

15. Sinha N. K.: Short Term Rheology of Polycrystalline Ice, J. Glaciol., 21(85), 457-473, 1978.

16. Sinha N.K.: Intercrystalline cracking, grain-boundary sliding, and delayed elasticity at high temperatures, J. Mat. Sci., 19, 359 – 376, 1984.

17. Sinha N.K.: Creep cracking and Acoustic Emission in polar shelf ice at high temperature of 0.96 $T_m$, Proc. 14[th] World Conference on Non-Destructive Testing (14[th] WCNDT), New Delhi (December 8-13 1996), 2475-2478, 1996.

18. Sommerfeld R. A. and Lachapelle E.: The Classification of snow metamorphism, J. of Glaciol. 9 (55), 1970.

19. Weiss J., Lahaie F., Grasso J. R.: Statistical analysis of dislocation dynamics during viscoplastic deformation from acoustic emission, J. of Geophy. Res. 105(B1)) 433-442, 2000.